# Ancestral protein reconstruction reveals evolutionary events governing variation in Dicer helicase function

**Adedeji M Aderounmu[1], P Joseph Aruscavage[1], Bryan Kolaczkowski[2], Brenda L Bass[1]***

[1]Department of Biochemistry, University of Utah, Salt Lake City, United States; [2]Department of Microbiology and Cell Science, University of Florida, Gainesville, United States

**Abstract** Antiviral defense in ecdysozoan invertebrates requires Dicer with a helicase domain capable of ATP hydrolysis. But despite well-conserved ATPase motifs, human Dicer is incapable of ATP hydrolysis, consistent with a muted role in antiviral defense. To investigate this enigma, we used ancestral protein reconstruction to resurrect Dicer's helicase in animals and trace the evolutionary trajectory of ATP hydrolysis. Biochemical assays indicated ancient Dicer possessed ATPase function, that like extant invertebrate Dicers, is stimulated by dsRNA. Analyses revealed that dsRNA stimulates ATPase activity by increasing ATP affinity, reflected in Michaelis constants. Deuterostome Dicer-1 ancestor, while exhibiting lower dsRNA affinity, retained some ATPase activity; importantly, ATPase activity was undetectable in the vertebrate Dicer-1 ancestor, which had even lower dsRNA affinity. Reverting residues in the ATP hydrolysis pocket was insufficient to rescue hydrolysis, but additional substitutions distant from the pocket rescued vertebrate Dicer-1's ATPase function. Our work suggests Dicer lost ATPase function in the vertebrate ancestor due to loss of ATP affinity, involving motifs distant from the active site, important for coupling dsRNA binding to the active conformation. By competing with Dicer for viral dsRNA, RIG-I-like receptors important for interferon signaling may have allowed or actively caused loss of ATPase function.

*For correspondence:
bbass@biochem.utah.edu

Competing interest: The authors declare that no competing interests exist.

## Editor's evaluation

This is a valuable paper describing an attempt to reconstruct the evolution of Dicer. Using ancestral reconstruction approaches, the authors carefully examine the biochem ical characteristics of reconstructed proteins at various junction points in the animal lineage. The authors provide solid evidence that the deepest ancestrally reconstructed protein has double-stranded RNA stimulated ATPase activity and that this characteristic was lost along the vertebrate lineage. This paper will be of interest to scientists in the RNA-protein interaction and protein evolution fields.

## Introduction

Dicer is a multidomain endoribonuclease that is conserved in most eukaryotes (*Jia et al., 2017*; *Gao et al., 2014*; *Fukudome and Fukuhara, 2017*; *Kidwell et al., 2014*; *Tabara et al., 2021*; *Bernstein et al., 2001*). Some organisms encode only a single Dicer (*Bernstein et al., 2001*; *Kim et al., 2009*; *Ha and Kim, 2014*; *Watanabe et al., 2008*; *Li et al., 2016*; *Qiu et al., 2017*; *Ashe et al., 2013*; *Welker et al., 2010*), while others encode multiple Dicers (*Fukudome and Fukuhara, 2017*; *Kidwell et al., 2014*; *Lee et al., 2004*; *Cenik et al., 2011*), with different versions specialized for pre-miRNA processing or endogenous/viral double-stranded (dsRNA) processing

(*Lee et al., 2004*; *Cenik et al., 2011*; *Wang et al., 2021*; *Wei et al., 2021*; *Deleris et al., 2006*; *Loffer et al., 2022*; *Poirier et al., 2018*; *Deddouche et al., 2008*; *Tsutsumi et al., 2011*; *Wu et al., 2020*). Dicer contains an intramolecular dimer of two RNaseIII domains, the catalytic center that cleaves dsRNA. It also contains a platform/PAZ domain, an N-terminal helicase domain, a C-terminal dsRNA-binding motif (dsRBM), and a domain of unknown function (DUF283) with a degenerate dsRBM fold (*Bernstein et al., 2001*; *Wei et al., 2021*; *Liu et al., 2018*; *Figure 1A*). These domains mediate recognition, binding, and discrimination of different dsRNAs, ensuring that optimal Dicer substrates are presented to the catalytic center for cleavage. The size of the small RNA product, either miRNA or siRNA, is defined by the distance between the platform/PAZ domain, which binds the ends of dsRNAs, and the RNaseIII domains (*Wang et al., 2021*; *Park et al., 2011*; *Gu et al., 2012*). Like the platform/PAZ domain, Dicer's helicase domain is capable of binding dsRNA termini, and in some organisms, the C-terminal dsRBM contributes to substrate binding and cleavage (*Wang et al., 2021*; *Su et al., 2022*; *Sinha et al., 2018*). Because both domains bind dsRNA termini, there is potential for the platform/PAZ and the helicase to compete for dsRNA substrates. To resolve this conflict, some extant metazoan Dicers have evolved substrate preferences where the platform/PAZ domain is specialized for binding the 2-nucleotide (nt) 3' overhang (3'ovr) of a pre-miRNA, while the helicase prefers dsRNA with blunt (BLT) termini (*Welker et al., 2011*; *Sinha et al., 2015*).

The role of the helicase domain varies among different Dicers. In *Drosophila melanogaster*, Dicer-1 (dmDcr1) specializes in pre-miRNA processing, but helicase function is not required (*Jiang et al., 2005*) while *D. melanogaster* Dicer-2 (dmDcr2), the second Dicer in fruit flies, uses its helicase domain to recognize and bind viral and long endogenous dsRNAs (*Wei et al., 2021*; *Su et al., 2022*; *Sinha et al., 2018*; *Singh et al., 2021*; *Naganuma et al., 2021*; *Trettin et al., 2017*; *Figure 1B*). Once bound, these dsRNAs are threaded by the helicase domain to the RNaseIII sites, using the energy of ATP hydrolysis for processive cleavage into siRNA products. This processive mechanism ensures that multiple siRNAs are produced from a single dsRNA (*Su et al., 2022*; *Singh et al., 2021*). Another invertebrate Dicer, *Caenorhabditis elegans* Dicer-1 (ceDCR-1), likewise requires a functional helicase domain to process long endogenous/viral dsRNAs, but like dmDcr1, it also processes pre-miRNA without a requirement for ATP (*Welker et al., 2010*; *Welker et al., 2011*). In contrast, *Homo sapiens* Dicer (hsDcr) has only been found to function in an ATP-independent manner, using its platform/PAZ domain to bind pre-miRNAs which are then distributively cleaved into mature miRNAs (*Figure 1C*; *Liu et al., 2018*; *Zhang et al., 2002*). Accordingly, the role of the single mammalian Dicer enzyme in anti-viral defense is controversial, as sensing of cytosolic viral dsRNAs is primarily mediated by RIG-I-Like receptors (RLRs), a family of enzymes that contain a related helicase domain (*Loo and Gale, 2011*; *Goubau et al., 2013*; *Ahmad and Hur, 2015*; *Luo et al., 2011*; *van der Veen et al., 2018*). Pathogenic dsRNA recognition by RLRs leads to production of interferon, which in turn triggers production of multiple antiviral proteins to suppress viral replication (*Loo and Gale, 2011*; *Rehwinkel and Gack, 2020*). Thus, helicase function in invertebrate Dicers correlates with a role in antiviral defense that seems to have been replaced by RLRs in mammals.

To understand the biochemical basis of the functional diversity between Dicer helicase domains, and by inference, their roles in antiviral defense, we used phylogenetics to reconstruct evolution of Dicer's helicase domain in animals. We included DUF283 in our analyses as recent Dicer structures reveal its role in binding dsRNA in concert with the helicase domain (*Wei et al., 2021*; *Su et al., 2022*). Ancestral protein reconstruction (APR) generates hypothetical protein sequences that are assumed to be reasonable approximations of real proteins that existed at different points in the distant past. Inherent uncertainty in the accuracy of sequence predictions and limited statistical power in single gene/protein sequences limit the methodology, but in spite of these limitations, APR offers a powerful tool for understanding the evolutionary journey that led to variation in extant proteins. Combining phylogenetic tree construction, APR, and biochemical analyses of reconstructed proteins, we show that basal and dsRNA-dependent ATPase function was present in ancestral animal Dicer. This capability declined between early animal evolution and the origin of deuterostomes and is lost entirely at the onset of vertebrate evolution. We also find that ancient Dicer helicases generally bind dsRNA more tightly than more modern and extant Dicer helicase domains. dsRNA binding to ancestral Dicer helicases stimulates ATP hydrolysis primarily by increasing the helicase domain's affinity for ATP, as reflected in differences in $K_M$ values observed in the absence and presence of dsRNA. Finally, we

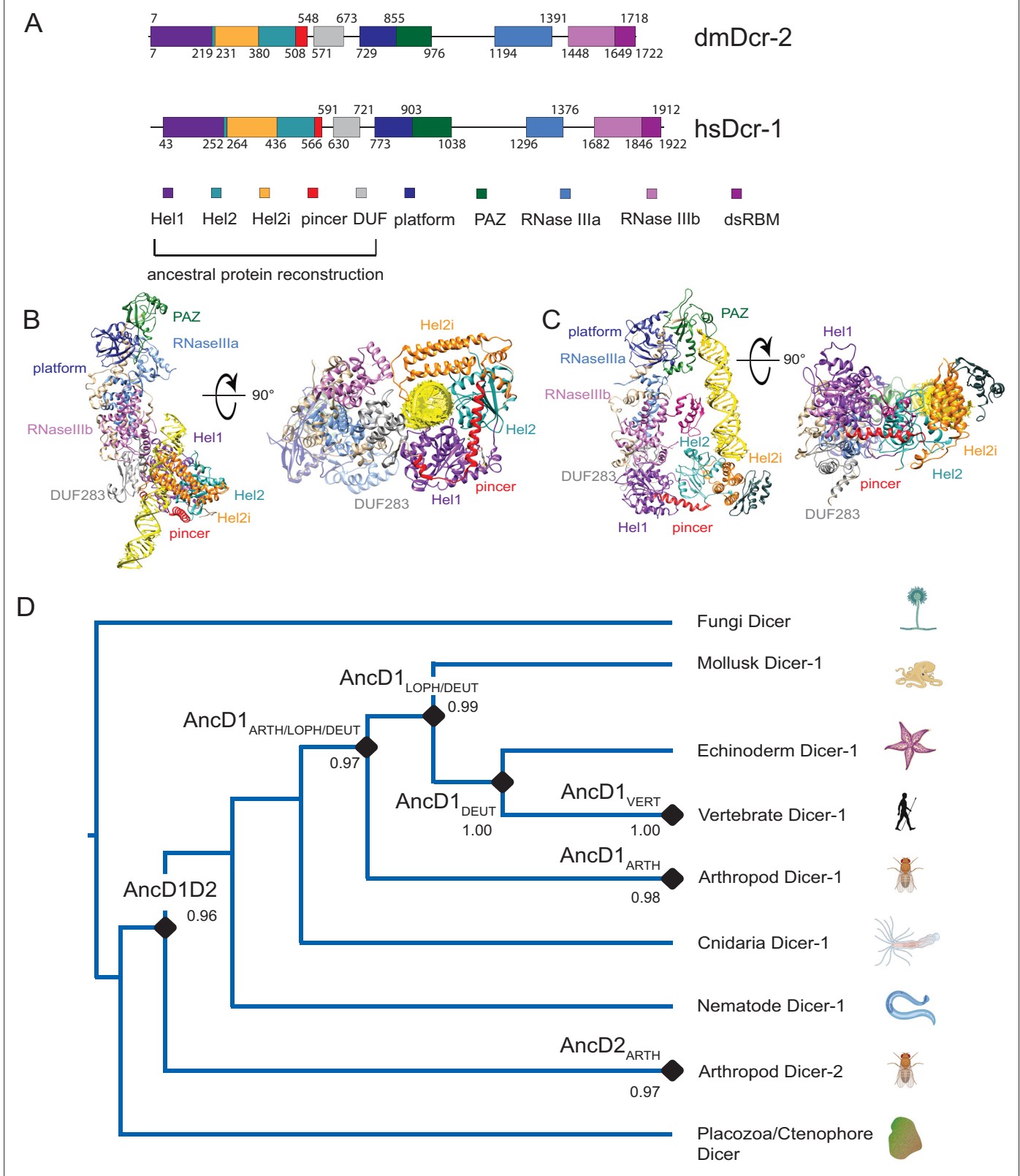

**Figure 1.** Phylogenetic analysis of Helicase domains and DUF283 of metazoan Dicer proteins. (**A**) Domain organization of *Drosophila melanogaster* Dicer-2 (dmDcr2) and *Homo sapiens* Dicer (hsDcr), with colored rectangles showing conserved domain boundaries indicated by amino acid number. Domain boundaries were defined by information from NCBI Conserved Domains Database (CDD), available crystal and cryo-EM structures, and structure-based alignments from previous studies (*Liu et al., 2018*; *Sinha et al., 2018*; *Marchler-Bauer et al., 2007*). (**B**) Structure of dmDcr2 bound

*Figure 1 continued on next page*

Figure 1 continued

to dsRNA (yellow) at the helicase domain. Left: front view. Right: bottom-up view. (PDB: 7W0C). (**C**) Structure of hsDcr bound to dsRNA (yellow) at the platform/PAZ domain. Left: front view. Right: bottom-up view. (PDB: 5ZAL). (**D**) Summarized maximum likelihood phylogenetic tree constructed from metazoan Dicer helicase domains and DUF283. Nodes of interest are indicated with black rounded rhombi, and transfer bootstrap values are indicated. ARTH: arthropod, LOPH: lophotrochozoa, DEUT: deuterostome, and VERT: vertebrate.

The online version of this article includes the following source data and figure supplement(s) for figure 1:

**Figure supplement 1.** Maximum likelihood phylogenetic tree constructed from metazoan Dicer helicase domains and DUF283.

**Figure supplement 2.** Alternative reconstructions of phylogenetic tree depicting Dicer HEL-DUF evolution.

**Figure supplement 3.** Constraining the phylogenetic tree to species-accurate relationships does not significantly impact ancestral protein reconstruction.

**Figure supplement 4.** Reconstructed HEL-DUF constructs are predicted with high confidence and expressed recombinantly.

**Figure supplement 4—source data 1.** Original digital image of SDS-PAGE gel used in B.

**Figure supplement 4—source data 2.** Posterior probabilities for ancestral states for all ancestrally reconstructed nodes in the maximum likelihood phylogeny.

partially resurrect ATPase function in the ancestral vertebrate Dicer helicase domain and find that loss of ATPase function is driven by amino acid substitutions distant from the catalytic pocket.

## Results

### Phylogenetic analysis of Dicer's helicase domain reveals an ancient gene duplication of animal Dicer

Dicer's large size (~220 kDa) and the significant sequence divergence between its homologs and paralogs (e.g. ~25% identity between hsDcr and dmDcr2) introduce uncertainty into multiple sequence alignments (MSAs), phylogeny construction, and ancestral protein resurrection (*Jia et al., 2017*; *Sinha et al., 2018*). Here, we focused our phylogenetic analyses on the helicase domain and DUF283 (HEL-DUF), two domains involved in functional diversity across metazoan Dicers (*Figure 1A*). Animal Dicer sequences were retrieved from NCBI databases and truncated to leave the helicase domain and DUF283. We used the HEL-DUF sequence alignment to infer a maximum likelihood (ML) phylogenetic tree and reconstructed the ancestral amino acid sequence on nodes from this phylogeny. The ML tree displayed an early gene duplication event for HEL-DUF, where an ancestral animal Dicer (AncD1D2) was split into two major Dicer clades, AncD1 and AncD2 (*Figure 1D*, *Figure 1—figure supplement 1*). AncD1 contains the ubiquitous Dicer-1 found in most animal species, while AncD2 contains the arthropod-specific Dicer-2. The observed gene duplication is consistent with previously reported phylogenetic analyses of full-length Dicer, suggesting that the HEL-DUF region contained enough phylogenetic signal to recapitulate broad patterns of Dicer evolution (*Jia et al., 2017*; *Gao et al., 2014*; *Mukherjee et al., 2013*).

Importantly, we observed multiple instances where our ML tree was not congruent with the species tree. Thus, while AncD1D2 corresponds to a singular Dicer ancestor that existed early in metazoan evolution, our tree does not resolve early events in animal evolution well enough to determine the exact timing of AncD1D2 duplication during the evolution of non-bilaterian animals. We observed several incongruences in the phylogeny that could stem from long branch attraction artifacts from the Dicer-2 clade, which has undergone rapid evolution under the selection pressure of viral defense and may not have diverged as early as it appears in the phylogeny (*Kolaczkowski et al., 2011*). Previous work attempting to determine the likelihood of an alternative phylogeny where the Dicer-2 clade arises from a more recent arthropod-specific duplication found it significantly less likely than the early gene duplication model (*Jia et al., 2017*; *Mukherjee et al., 2013*), but this conclusion may be due to a limitation of existing statistical tools.

To understand how changes in the phylogeny affect the predicted ancestral sequence, we constrained the ML tree to known bilaterian species relationships (*Figure 1—figure supplement 2A*), resulting in a few changes in the reconstructed sequences for AncD1D2 and AncD1$_{DEUT}$. However, the variable amino acids are not predicted to significantly affect the observed biochemical properties (96% identity between ML tree and species tree reconstruction; *Figure 1—figure supplement 3A, B*), and APR performed using the species tree did not alter the AncD1$_{VERT}$ amino acid sequence. This version

of the tree also indicates that AncD1D2 is ancestral to the bilaterian ancestor, AncD1$_{BILAT}$. Finally, constraining the arthropod Dicer-2 clade in the species tree to a recent arthropod-specific duplication event produces a more parsimonious tree with worse likelihood scores, with a log-likelihood score of –435,841 compared to –435,706 for the ML tree (*Figure 1—figure supplement 2B*; *Jia et al., 2017*).

We proceeded to use the HEL-DUF ML tree for APR as the information from these hypothetical reconstructed ancestral nodes proved valuable for understanding the processes that underpin loss of helicase function in hsDcr. Select nodes in the ML tree were subjected to APR using RAXML-NG. Amino acid sequences for AncD1D2, AncD2$_{ARTH}$, AncD1$_{ARTH/LOPH/DEUT}$, AncD1$_{LOPH/DEUT}$, AncD1$_{DEUT}$, and AncD1$_{VERT}$ were predicted with a high degree of confidence. AncD1$_{VERT}$ had more than 95% of sites with posterior probabilities of 0.8 or above, while older constructs had an average of 75% of sites with posterior probabilities above 0.8 (*Figure 1—figure supplement 4A*). Ancestral constructs were expressed recombinantly using baculovirus in Sf9 insect cells and purified to 99% homogeneity (*Figure 1—figure supplement 4B*; *Sinha and Bass, 2017*). Protein identity was confirmed with LC/MS/MS.

## Ancient animal Dicer helicase domain possessed dsRNA-stimulated ATPase function

Certain extant Dicer enzymes have ATP hydrolysis activity, while others do not, suggesting either a gain or loss of this activity during evolution. To understand the source of this variation, we performed multiple-turnover ATP hydrolysis assays of ancestral HEL-DUF proteins with and without dsRNA. We observed ATP hydrolysis in the most recent common ancestor of hsDcr and dmDcr2, AncD1D2 (*Figure 2A*), leading to the important conclusion that extant animal Dicers with no dependence on ATP, such as hsDcr, lost the capacity for ATP hydrolysis subsequent to this period in animal evolution. Basal ATP hydrolysis activity was present at low levels in AncD1D2 ($k_{obs}$: 0.06 min$^{-1}$) and was improved upon addition of dsRNA (*Figure 2A*, *Table 1*).

Adding dsRNA to AncD1D2 showed a dramatic increase in the ADP produced over time (*Figure 2A*, right panel). To enable better comparison of efficient HEL-DUF ancestors and minimize the effects of substrate depletion during the reaction, data were modeled as a two-phase exponential curve. The first phase was represented by a fast linear burst of ATP hydrolysis capturing a transient zero order reaction where rate is independent of ATP concentration, and the second phase was modeled as a slow, first order exponential increase in ATP hydrolysis, where, because of robust hydrolysis, ATP concentration falls below some affinity threshold for the enzyme (*Figure 2E and F*, *Table 1*; *Tóth et al., 2015*). Without dsRNA, ATP hydrolysis is a slow first order reaction because the concentration of ATP in this reaction (100 µM) is likely orders of magnitude below the affinity threshold. Addition of dsRNA with BLT termini to AncD1D2, a substrate designed to mimic termini of certain RNA viruses, promoted hydrolysis of ATP in the fast phase ($k_{burst}$: 14.3 µM/min) as well as doubling the rate constant of the slow phase ($k_{obs}$: 0.11 min$^{-1}$; *Table 1*). Similar rates were observed when a dsRNA with a 2-nucleotide (nt) 3' overhang (3'ovr; pre-miRNA mimic) was used (*Table 1*). Lack of dsRNA terminus discrimination suggests a substrate promiscuity that is absent in modern Dicers, where BLT dsRNA is the optimal substrate for the helicase domain (*Sinha et al., 2018*; *Welker et al., 2011*; *Sinha et al., 2015*; *Singh et al., 2021*).

The arthropod Dicer-2 ancestor, AncD2$_{ARTH}$, was more efficient at hydrolyzing ATP in the absence of dsRNA than AncD1D2 and all other ancestors tested (*Figure 2B and E*), showing a two-phase reaction resembling a dsRNA-stimulated reaction ($k_{burst}$: 6.47 µM/min, $k_{obs}$: 0.05 min$^{-1}$; *Table 1*). This efficient hydrolysis in the absence of dsRNA suggests that AncD2$_{ARTH}$ refined its ATP binding pocket to produce high affinity for ATP even in the absence of nucleic acid. Addition of BLT dsRNA increased the rate of the fast phase ($k_{burst}$: 19.3 µM/min), with slightly better efficiency compared to 3'ovr dsRNA ($k_{burst}$:14.6 µM/min; *Table 1*, *Figure 2B and F*), perhaps foreshadowing the terminus discrimination seen in modern dmDcr2 (*Sinha et al., 2015*; *Singh et al., 2021*). Interestingly, terminus discrimination was also observed in Dicer-1 ancestors, suggesting the foundation of this discrimination existed in AncD1D2, even if not observable at the conditions tested. The deuterostome Dicer-1 node immediately preceding vertebrate Dicer-1, AncD1$_{DEUT}$, had a pattern of basal and dsRNA-stimulated ATP hydrolysis similar to AncD1D2 (*Table 1*; *Figure 2C, E and F*) with a reduction in overall hydrolysis efficiency but the appearance of discrimination when triggered by different dsRNA termini (BLT, $k_{burst}$: 6.0 µM/min; 3'ovr, $k_{burst}$: 1.4 µM/min; *Table 1*). This reduction in ATP hydrolysis efficiency prior to

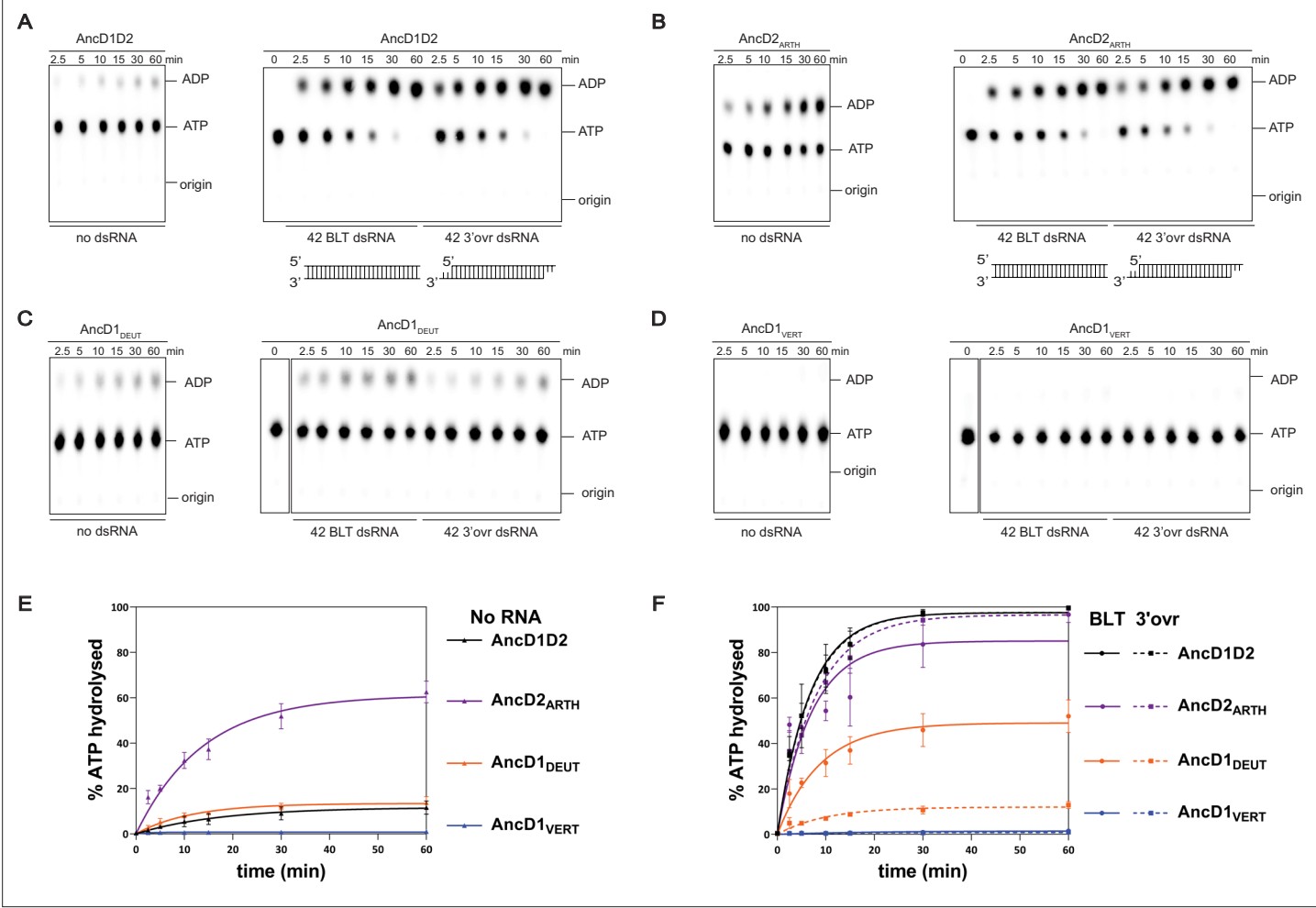

**Figure 2.** ATP hydrolysis capability is present in ancestral metazoan Dicer but lost at the common ancestor of vertebrates. (**A–D**) PhosphorImages of representative thin-layer chromatography (TLC) plates showing hydrolysis of 100 µM ATP (spiked with α-$^{32}$P-ATP) by 200 nM ancestral HEL-DUFs for various times as indicated, at 37°C, in the absence of dsRNA or in the presence of 400 nM 42 base-pair dsRNA with blunt (BLT) or 3'ovr termini (see cartoons; not radiolabeled). (**E**) Graph shows quantification of ATP hydrolysis assays (**A–D**) performed with select ancestral HEL-DUF enzymes in the absence of dsRNA. Data for 'NO RNA' reactions were fit to the pseudo-first order equation $y=y_o + A \times (1-e^{-kt})$; where y=product formed (ADP in µM); A=amplitude of the rate curve, $y_o$ = baseline (~0), k=pseudo-first-order rate constant = $k_{obs}$; t=time. Data points are mean ± SD (n≥3). (**F**) Graph shows quantification of ATP hydrolysis assays (**A–D**) performed with select ancestral HEL-DUF enzymes in the presence of dsRNA. Reactions with RNA were fit in two phases, first a linear phase for data below the first timepoint at 2.5 min, then a pseudo-first order exponential equation for remaining data. Equation, $y=y_o + A \times (1-e^{-kt})$; where y=product formed (ADP in µM); A=amplitude of the rate curve, $y_o$ = baseline (~0), k=pseudo-first-order rate constant = $k_{obs}$; t=time. Data points are mean ± SD (n≥3).

The online version of this article includes the following source data for figure 2:

**Source data 1.** Raw digital images of thin-layer chromatography plate used in 2A.

**Source data 2.** Raw digital image of thin-layer chromatography plate used in 2A.

**Source data 3.** Raw digital image of thin-layer chromatography plate used in 2B.

**Source data 4.** Raw digital image of thin-layer chromatography plate used in 2B.

**Source data 5.** Raw digital image of thin-layer chromatography plate used in 2C.

**Source data 6.** Raw digital image of thin-layer chromatography plate used in 2C.

**Source data 7.** Raw digital image of thin-layer chromatography plate used in 2D.

**Source data 8.** Raw digital image of thin-layer chromatography plate used in 2D.

**Table 1.** Summary of kinetic data for ATP hydrolysis with 100 µM ATP.

| Construct | $k_{burst}$ (µM/min), NO RNA<br>$k_{obs}$ (min$^{-1}$) | $k_{burst}$ (µM/min), BLT dsRNA<br>$k_{obs}$ (min$^{-1}$) | $k_{burst}$ (µM/min), 3'ovr dsRNA<br>$k_{obs}$ (min$^{-1}$) |
|---|---|---|---|
| AncD1D2 | -<br>0.06±0.01 | 14.3±1.7<br>0.11±0.03 | 13.9±0.5<br>0.11±0.02 |
| AncD2$_{ARTH}$ | 6.47±0.8<br>0.05±0.01 | 19.3±0.9<br>0.04±0.02 | 14.6±2.3<br>0.08±0.02 |
| AncD1$_{ARTH/LOPH/DEUT}$ | -<br>0.01±0.01 | 25.1±0.7<br>0.41±0.02 | 16.0±3.8<br>0.21±0.04 |
| AncD1$_{LOPH/DEUT}$ | -<br>0.07±0.02 | 7.4±0.3<br>0.04±0.01 | 3.8±0.6<br>0.03±0.01 |
| AncD1$_{DEUT}$ | -<br>0.09±0.02 | 6.0±0.7<br>0.06±0.03 | 1.4±0.03<br>0.06±0.02 |
| AncD1$_{VERT}$ | - | - | - |

deuterostome divergence may have coincided with, or may have been caused by, the appearance of innate immune effector proteins in invertebrate evolution.

Most importantly, in the ancestor of vertebrate Dicer, AncD1$_{VERT}$, ATP hydrolysis was undetectable (*Figure 2D*), consistent with observations that modern hsDcr is incapable of ATP hydrolysis (*Zhang et al., 2002*), and indicating that loss of Dicer ATPase activity progressed further between the deuterostome ancestor and the vertebrate ancestor, probably driven by a collection of evolutionary events. While an ancestral protein-based innate immune system existed prior to deuterostomes, the appearance of the interferon molecule and its expanded role in innate and adaptive immunity may have contributed to further loss of Dicer helicase function and set the stage for miRNA expansion (*Dehal and Boore, 2005*; *Secombes and Zou, 2017*; *Qiao et al., 2021*; *Berezikov, 2011*).

## Ancestral HEL-DUF binds dsRNA with higher affinity than modern Dicer HEL-DUF

Our experiments indicated that dsRNA improves ATP hydrolysis by ancient Dicer helicase domains, in some cases, in a terminus-dependent manner. We wondered if terminus discrimination occurred during initial dsRNA binding. To investigate the dsRNA•HEL-DUF interaction, as well as how it is affected by ATP, we used electrophoretic mobility shift assays (EMSAs) with HEL-DUFs to measure the dissociation constant ($K_d$) in the presence and absence of ATP, using BLT or 3'ovr dsRNA (*Figure 3A*; *Hellman and Fried, 2007*; *Rio, 2014*).

All ancestral proteins bound dsRNA and showed multiple shifts that typically decreased in mobility with increasing protein concentration. AncD1D2, the most ancient construct tested, displayed tight binding to BLT dsRNA without ATP ($K_d$, 3.4 nM), while the addition of 5 mM ATP caused a twofold reduction in affinity ($K_d$, 6.4 nM; *Table 2*, *Figure 3B and H*). Binding to 3'ovr dsRNA was similarly tight, albeit with an ~twofold reduction in binding affinity compared to BLT in the absence or presence of ATP (*Table 2*, *Figure 3C and H*). This suggests that dsRNA binding is the earliest point of terminus discrimination for ATP hydrolysis with ancestral HEL-DUFs. Possibly, the observed lower affinity with ATP occurs because ATP hydrolysis promotes dissociation or translocation that results in HEL-DUF sliding off dsRNA.

An obvious feature evident from the binding isotherms (*Figure 3H*) was that the more ancient the HEL-DUF construct in our tree, the tighter its interaction with dsRNA, regardless of the presence or absence of ATP. Comparison of $K_d$ values (*Table 2*) revealed other interesting trends. AncD1$_{DEUT}$, the common ancestor of deuterostomes, which include humans, had a lower binding affinity for dsRNA compared to AncD1D2, regardless of termini or the presence of ATP (~10–50-fold, *Table 2*, *Figure 3D and H*). In addition, the ability to distinguish BLT and 3'ovr dsRNA largely disappeared, and ATP had little effect on dsRNA binding (*Table 2*, *Figure 3E and H*). This absence of discrimination between termini or ATP-bound states stands in contrast to observations for AncD1D2 and extant *D. melanogaster*, whose helicase domains bind BLT dsRNA better than 3'ovr dsRNA (*Sinha et al., 2018*; *Naganuma et al., 2021*). This lack of discrimination in binding does not match the sensitivity

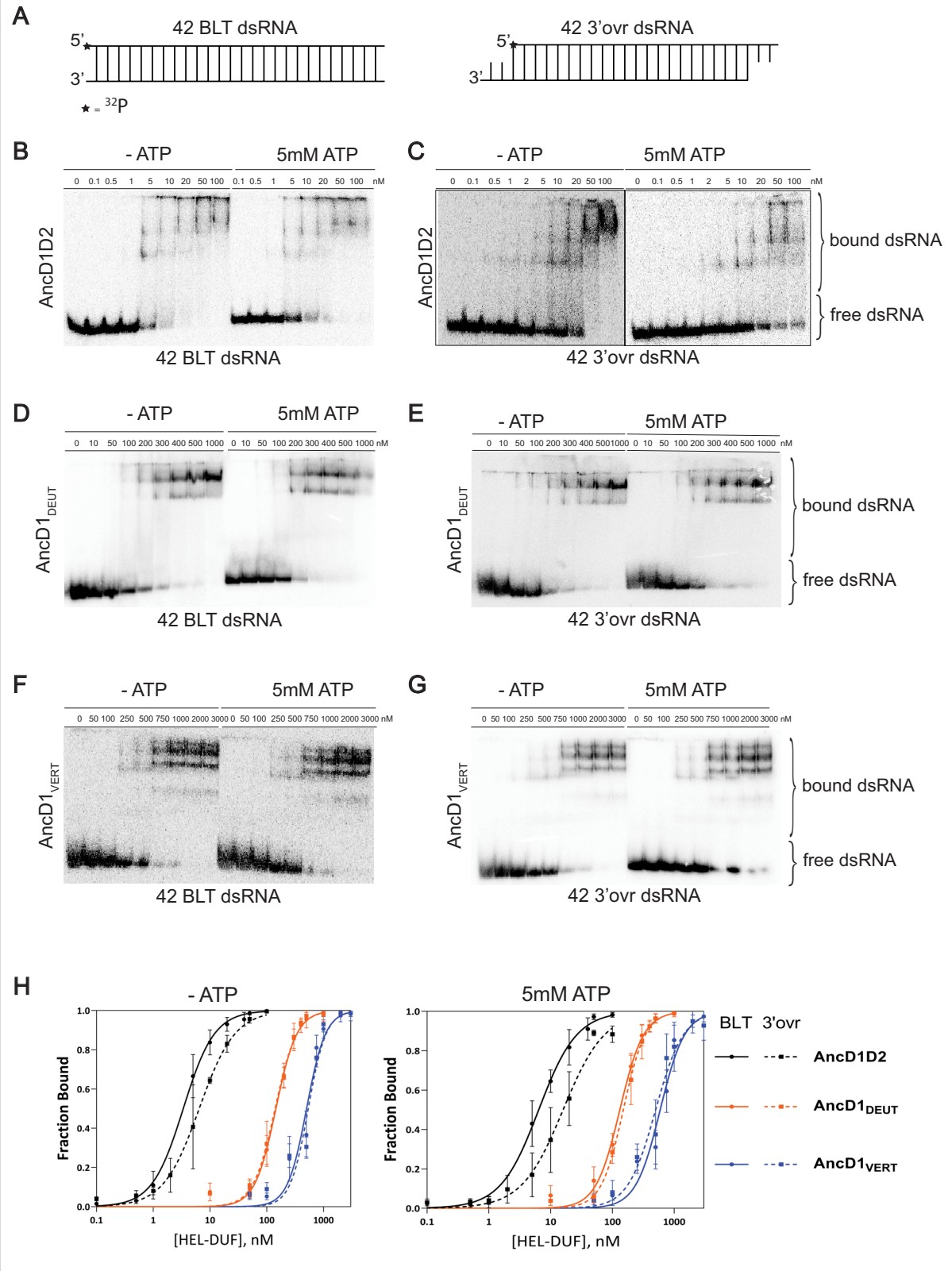

**Figure 3.** Binding affinity of ancestral HEL-DUF proteins to blunt (BLT) and 3'ovr dsRNA in the presence and absence of ATP. (**A**) Cartoon of dsRNAs used in (**B–G**) showing position of 5' $^{32}$P (*) on top, sense strand. (**B–G**) Representative PhosphorImages showing gel mobility shift assays using select ancestral HEL-DUF constructs, as indicated, and 42-base pair BLT or 3'ovr dsRNA in the absence (-) or presence of 5 mM ATP at 4°C. (**H**) Radioactivity in PhosphorImages as in A–G was quantified to generate binding isotherms for ancestral HEL-DUF proteins. Fraction bound was determined using

*Figure 3 continued on next page*

*Figure 3 continued*

radioactivity for dsRNA$_{free}$ and dsRNA$_{bound}$. Data were fit to calculate dissociation constant, $K_d$, using the Hill formalism, where fraction bound = $1/(1 + [K_d^n/[P]^n])$. Data points, mean ± SD (n≥3).

The online version of this article includes the following source data for figure 3:

**Source data 1.** Raw digital image of Gel Shift phosphoimager plate used in 3B.

**Source data 2.** Raw digital image of Gel Shift phosphoimager plate used in 3C.

**Source data 3.** Raw digital image of Gel Shift phosphoimager plate used in 3C.

**Source data 4.** Raw digital image of Gel Shift phosphoimager plate used in 3D.

**Source data 5.** Raw digital image of Gel Shift phosphoimager plate used in 3E.

**Source data 6.** Raw digital image of Gel Shift phosphoimager plate used in 3F.

**Source data 7.** Raw digital image of Gel Shift phosphoimager plate used in 3G.

to dsRNA termini observed in ATP hydrolysis (*Figure 2C*), suggesting an additional discriminatory step exists between initial dsRNA binding and ATP hydrolysis. Another possibility is that for some constructs but not for others, ATP's effect on dsRNA binding is muted or altered at the low temperatures (4°C) where EMSAs were performed.

Binding of dsRNA to AncD1$_{VERT}$ resembled binding to AncD1$_{DEUT}$ in that affinity did not depend on termini or ATP (*Table 2*, *Figure 3F, G and H*). However, AncD1$_{VERT}$ binding to dsRNA was weaker across all conditions with ~4.5-fold reduction in affinity compared to AncD1$_{DEUT}$ and ~30–150-fold reduction compared to AncD1D2 (*Table 2*). Interestingly, the $K_d$ values measured for AncD1$_{VERT}$ were similar to values reported for the modern hsDcr helicase domain, or hsDcr with the platform/PAZ domain mutated to abolish competing binding, hinting that this construct correlates well with extant biology (*Tian et al., 2014*; *Ma et al., 2012*). One possibility is that between AncD1D2 and the deuterostome ancestor, AncD1$_{DEUT}$, HEL-DUF's dsRNA affinity decreased as the platform/PAZ domain began to play a more significant role in binding 3′ovr termini of pre-miRNAs, and RLRs co-opted binding of virus-like BLT dsRNAs. In summary, the more ancient the HEL-DUF construct in our tree, the tighter the dsRNA•HEL-DUF interaction, with the deuterostome HEL-DUF ancestor losing the ability to discriminate dsRNA termini by binding.

## BLT dsRNA improves ATP hydrolysis by improving the association of ATP to HEL-DUF

Our analyses so far showed that dsRNA markedly altered the kinetics and efficiency of ATP hydrolysis. To understand how dsRNA binding affected the interaction of the helicase with ATP, we performed Michaelis-Menten analyses. We focused on determining kinetics for ATP hydrolysis catalyzed by AncD1D2 and AncD1$_{DEUT}$ to gain information about two Dicer-1 ancestors that presumably correspond to different periods in Dicer evolution. Without added dsRNA, basal ATP hydrolysis for AncD1D2

**Table 2.** Dissociation constants for dsRNA binding to ancestral HEL-DUFs.

| Construct | $K_d$ (nM) BLT, NO ATP Hill coefficient | $K_d$ (nM) 3′ovr, NO ATP Hill coefficient | $K_d$ (nM) BLT, 5 mM ATP Hill coefficient | $K_d$ (nM) 3′ovr, 5 mM ATP Hill coefficient |
|---|---|---|---|---|
| AncD1D2 | 3.4±0.4 1.6±0.2 | 6.5±0.8 1.4±0.2 | 6.4±0.7 1.4±0.2 | 15.9±2.4 1.3±0.2 |
| AncD2$_{ARTH}$ | n.d. | n.d. | n.d. | n.d. |
| AncD1$_{ARTH/LOPH/DEUT}$ | 23.8±2.2 2.0±0.4 | 40.1±3.7 1.7±0.3 | 17.5±2.1 1.6±0.3 | 17.3±1.5 1.7±0.2 |
| AncD1$_{LOPH/DEUT}$ | 60.9±5.9 1.9±0.3 | 90.8±8.8 1.4±0.2 | 38.0±3.5 2.3±0.5 | 49.0±5.2 1.4±0.2 |
| AncD1$_{DEUT}$ | 145.1±9.1 2.3±0.3 | 140.0±8.9 2.3±0.3 | 131.8±8.7 2.2±0.3 | 149.8±8.3 2.4±0.3 |
| AncD1$_{VERT}$ | 502.4±40.5 2.6±0.5 | 537.8±47.5 2.8±0.6 | 592±48.6 2.2±0.4 | 500.3±58.0 1.9±0.4 |

**Table 3.** Michaelis-Menten parameters for steady state ATP hydrolysis reactions.

| Construct | $k_{cat}$ (min$^{-1}$) | $K_M$ (µM) | $k_{cat}/K_M$ (µM$^{-1}$ min$^{-1}$) |
|---|---|---|---|
| AncD1D2, no dsRNA | 1117±94.5 | 35812±6,367 | 0.031 |
| AncD1D2, BLT dsRNA | 147.8±6.3 | 256±67.5 | 0.577 |
| AncD1$_{DEUT}$, no dsRNA | 144.1±14.9 | 2550±855 | 0.055 |
| AncD1$_{DEUT}$, BLT dsRNA | 40.31±3.78 | 336.4±142 | 0.12 |
| AncD1$_{VERT}$, no dsRNA | - | - | - |
| AncD1$_{VERT.7}$, no dsRNA | 257.7±28.7 | 61739±12,886 | 0.004 |
| AncD1$_{VERT.7}$, BLT dsRNA | 24.87±3.52 | 5173±1929 | 0.005 |

had a $k_{cat}$ of 1117 min$^{-1}$ and a $K_M$ of 35.8 mM (**Table 3**, **Figure 4A**, **Figure 4—figure supplement 1**). Adding excess BLT dsRNA caused $k_{cat}$ to drop ~eightfold to 147.8 min$^{-1}$ while $K_M$ dropped ~140-fold to 0.26 mM, indicating that although binding of BLT dsRNA to the AncD1D2 caused a reduction in the ATP turnover efficiency, it concurrently triggered a tighter association with ATP, leading to ~19-fold net improvement in $k_{cat}/K_M$ (**Table 3**). This improvement in efficiency was primarily evident at lower ATP concentrations that fall in the range of intracellular ATP concentrations (**Patel et al., 2017**; **Figure 4A**, right panel). These observations also explained the appearance of the fast linear phase in the multiple-turnover hydrolysis reactions performed with 100 µM ATP (**Table 1**, **Figure 2**). At this 'lower' ATP concentration, we speculate that dsRNA binding causes a conformational change in the helicase domain that allows tighter association with ATP, enabling a more efficient hydrolysis reaction, until ATP concentration falls below a threshold where the reaction slows. Without dsRNA, only slow ATP hydrolysis is available as the helicase rarely samples the conformations that allow tight interactions with ATP.

For the AncD1$_{DEUT}$ construct, basal hydrolysis proceeded with a $k_{cat}$ of 144.1 min$^{-1}$, ~eightfold lower than the rate recorded for the AncD1D2 reaction (**Table 3**, **Figure 4B**, **Figure 4—figure supplement 1C**). Association of ATP with the AncD1$_{DEUT}$ construct, as measured by $K_M$, was ~14-fold tighter leading to similar $k_{cat}/K_M$ values for both enzymes in the absence of dsRNA (**Table 3**). Adding BLT dsRNA caused a reduction in $k_{cat}$ by a factor of ~4, while reducing $K_M$ by a factor of ~8 to give a twofold net increase in $k_{cat}/K_M$ (**Table 3**, **Figure 4B**, **Figure 4—figure supplement 1D**). As with the AncD1D2 construct, improvement is primarily mediated by improved association of enzyme with ATP, evident at lower ATP concentrations (**Figure 4B**, right panel). Thus, we observed a trend where saturation of the helicase domain with BLT dsRNA caused improved ATP association to the catalytic ATPase site.

## Reversing historical substitutions in AncD1$_{VERT}$ partially rescued ATP hydrolysis

To acquire insight into the reason for loss of function in the ancestor of vertebrate Dicer, AncD1$_{VERT}$, we compared Michaelis-Menten kinetics of AncD1$_{VERT}$ and AncD1$_{DEUT}$. However, even at higher protein concentrations (5 µM), it was impossible to confidently discern whether AncD1$_{VERT}$ produced a signal over background using our thin-layer chromatography (TLC)-based assay (data not shown). Thus, we compared amino acid sequences of ancestral HEL-DUFs that retained ATPase activity to the sequence of the inactive vertebrate HEL-DUF ancestor and identified substitutions that might be responsible for the loss of activity (**Figure 4—figure supplement 2**). ATP hydrolysis and dsRNA-binding data for the ML tree-species tree incongruent nodes, AncD1$_{ARTH/LOPH/DEUT}$ and AncD1$_{LOPH/DEUT}$, proved useful here, allowing a deeper analysis of sequence-function relationships of Dicer's helicase domain (**Figure 4—figure supplements 3 and 4**). We created variants of AncD1$_{VERT}$, each with a subset of these substitutions, and purified two constructs, AncD1$_{VERT.1}$ and AncD1$_{VERT.7}$ (**Figure 4—figure supplement 2**). In AncD1$_{VERT.1}$, we reverted 20 amino acids which were either close to ATP binding/hydrolysis amino acid residues or to the ATPase active site in the tertiary structure, but this construct remained devoid of ATPase activity (data not shown). However, in AncD1$_{VERT.7}$, a construct containing an additional 21 amino acid substitutions distant from the catalytic site, ATPase activity was rescued, and we measured its Michaelis-Menten kinetics. Basal ATP hydrolysis had a $k_{cat}$ of 257.7 min$^{-1}$ and a $K_M$ of 61.7 mM (**Table 3**, **Figure 4C**, **Figure 4—figure supplement 1E**). Enzyme turnover was more efficient than

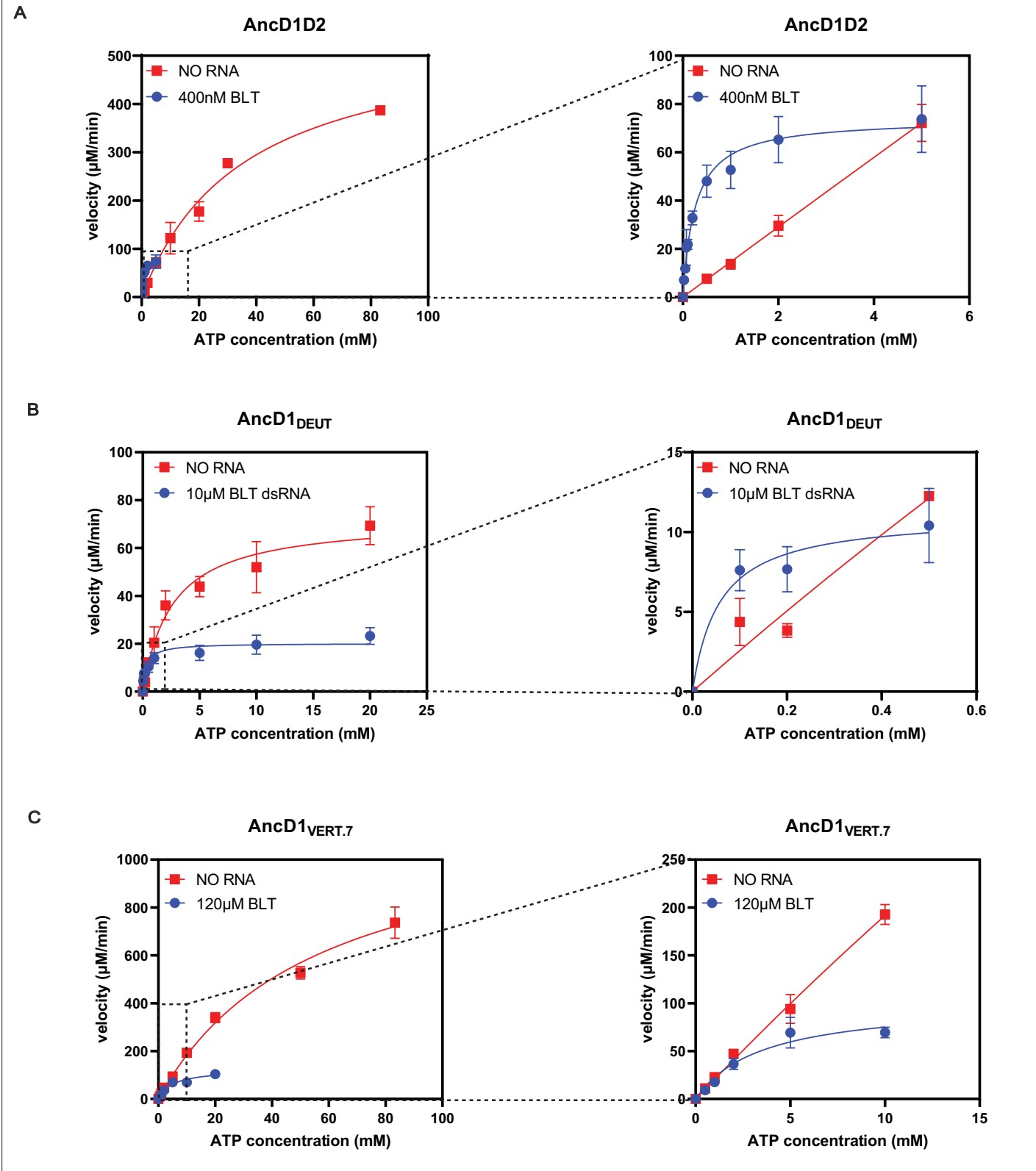

**Figure 4.** Blunt (BLT) dsRNA improves efficiency of ATP hydrolysis by improving affinity of ATP to ancient HEL-DUF enzymes. (**A**) Michaelis-Menten plots for basal and dsRNA-stimulated ATP hydrolysis by AncD1D2. Basal ATP hydrolysis measured at 500 nM AncD1D2, while dsRNA-stimulated hydrolysis is measured at 100 nM. Velocities for dsRNA-stimulated reaction have been multiplied by five to normalize this concentration difference. Right: inset showing Michaelis-Menten plot at low ATP concentrations. Hydrolysis data for individual ATP concentrations is included in *Figure 4—figure*

*Figure 4 continued on next page*

*Figure 4 continued*

*supplement 1*. (**B**) Michaelis-Menten plots for basal and dsRNA-stimulated ATP hydrolysis by 500 nM AncD1$_{DEUT}$. Right: inset showing Michaelis-Menten plot at low ATP concentrations. Hydrolysis data for individual ATP concentrations is included in *Figure 4—figure supplement 1*. (**C**) Michaelis-Menten plots for basal and dsRNA-stimulated ATP hydrolysis by 5 µM AncD1$_{VERT.7}$. Right: inset showing Michaelis-Menten plot at low ATP concentrations. Hydrolysis data for individual ATP concentrations is included in *Figure 4—figure supplement 1*. Data points, mean ± SD (n≥3).

The online version of this article includes the following source data and figure supplement(s) for figure 4:

**Figure supplement 1.** Plots of ADP production over time for ancestral HEL-DUF constructs.

**Figure supplement 2.** Multiple sequence alignment of ancestral HEL-DUF constructs and AncD1$_{VERT}$ rescue constructs.

**Figure supplement 3.** ATP hydrolysis of ancestral HEL-DUF proteins reconstructed from incongruent nodes.

**Figure supplement 3—source data 1.** Raw digital image of thin-layer chromatography plate used in *Figure 4—figure supplement 3A*, left panel.

**Figure supplement 3—source data 2.** Raw digital image of thin-layer chromatography plate used in *Figure 4—figure supplement 3A*, right panel.

**Figure supplement 3—source data 3.** Raw digital image of thin-layer chromatography plate used in *Figure 4—figure supplement 3B*.

**Figure supplement 3—source data 4.** Raw digital image of thin-layer chromatography plate used in *Figure 4—figure supplement 3B*.

**Figure supplement 4.** Affinity of AncD1$_{ARTH/LOPH/DEUT}$ and AncD1$_{LOPH/DEUT}$ for binding blunt (BLT) and 3'ovr dsRNA in the absence and presence of ATP.

**Figure supplement 4—source data 1.** Raw digital image of Gel Shift phosphoimager plate used in *Figure 4—figure supplement 4A*.

**Figure supplement 4—source data 2.** Raw digital image of Gel Shift phosphoimager plate used in *Figure 4—figure supplement 4B*.

**Figure supplement 4—source data 3.** Raw digital image of Gel Shift phosphoimager plate used in *Figure 4—figure supplement 4C*.

**Figure supplement 4—source data 4.** Raw digital image of Gel Shift phosphoimager plate used in *Figure 4—figure supplement 4D*.

AncD1$_{DEUT}$ but less efficient than AncD1D2, suggesting a rescue of the enzyme's inherent catalytic activity. However, the high K$_M$ value indicated that AncD1$_{VERT.7}$ was not rescued for a tight association with ATP, and the k$_{cat}$/K$_M$ value for this construct was ~10-fold lower than k$_{cat}$/K$_M$ for both AncD1$_{DEUT}$ and AncD1D2 (*Table 3*). Adding BLT dsRNA caused k$_{cat}$ to drop ~10-fold to 24.9 min$^{-1}$ while improving the K$_M$ by ~12-fold, therefore yielding no net improvement in k$_{cat}$/K$_M$. (*Table 3*, *Figure 4C*, *Figure 4—figure supplement 1F*). Our observations indicate that we have partially rescued ATPase activity in vertebrate HEL-DUF as well as the conformational changes that occur upon dsRNA binding. AncD-1$_{VERT.1}$, constructed by reversing candidate amino acid substitutions close in proximity to the conserved ATPase motifs, did not rescue ATPase activity. Instead, we found that amino acids distant from the ATP binding pocket were essential for resurrecting ATP hydrolysis in the vertebrate ancestor of Dicer.

## Changes in Dicer helicase domain's conformation influence ATPase activity

The effects of dsRNA binding on ancestral Dicer helicase domains show parallels to previously reported results for extant Dicers. In the absence of dsRNA, hsDcr and dmDcr2 helicase domains primarily exist in an 'open' conformation, with DUF283 wedged behind the Hel1 subdomain (*Figure 5A and B*; *Liu et al., 2018*; *Su et al., 2022*). dsRNA binding to dmDcr2's helicase domain causes a conformational change that brings DUF283 to the cleft of the helicase domain to clamp the dsRNA substrate (*Figure 5C*), while Hel2 and Hel2i shift their relative positions to create a 'closed' conformation. In the closed conformation, the distance between the 'DECH' box in Hel1 (Motif II) and the arginine finger motif (Motif VI) in Hel2 is reduced from 13.28 Å to 4.26 Å (*Figure 5D*; *Su et al., 2022*). In helicases formed by two RecA domains, the shorter distance is predicted to be bridged by water, the attacking nucleophile for cleavage of the gamma-phosphate of the ATP molecule (*Story and Steitz, 1992*; *Tian et al., 2016*). This change in conformation is consistent with the reduction in the K$_M$ values for ATP when BLT dsRNA is included in the ATP hydrolysis reaction for AncD1D2 and AncD1$_{DEUT}$ (*Figure 4*, *Table 3*). Along the same lines, a K$_M$ of 14 µM was reported for the dmDcr2 ATPase reaction in the presence of BLT dsRNA (*Cenik et al., 2011*). We predict that excluding dsRNA would also reduce dmDcr2's affinity for ATP, explaining why it does not hydrolyze ATP in the absence of dsRNA (*Sinha et al., 2015*).

AncD1$_{VERT}$ structures predicted by AlphaFold2 and RosettaFold have an open conformation, while AncD1D2 resembles the closed conformation (*Figure 5E*). All the other ATPase-competent ancestral HEL-DUFs also have a closed conformation (not shown). While these predictions are snapshots of a singular conformation from an ensemble of possible conformations, it is intriguing that the conformational differences between ancestral HEL-DUFs match our experimentally determined biochemical

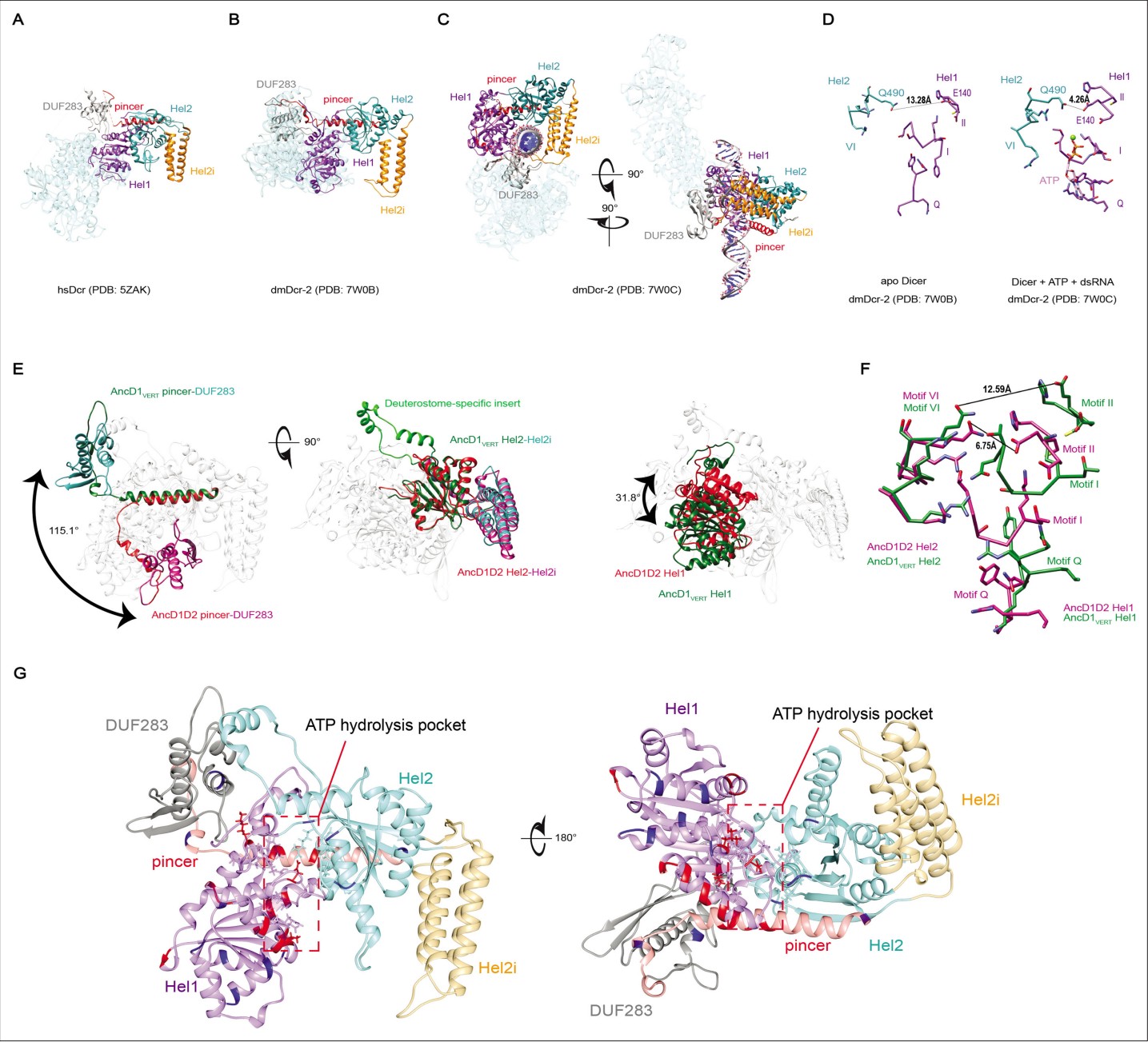

**Figure 5.** dsRNA binding triggers conformational changes in the HEL-DUF domains of Dicer. (**A**) Bottom-up view of the structure of *Homo sapiens* Dicer (hsDcr) in the apo state (PDB: 5ZAK). Helicase subdomains and DUF283 are colored. Rest of enzyme is transparent. (**B**) Bottom-up view of the structure of dmDcr2 in the apo state (PDB: 7W0B). Helicase subdomains and DUF283 are colored for visibility. Rest of enzyme is transparent. (**C**) Structure of dmDcr2 bound to dsRNA in the 'early translocation' state (PDB: 7W0C). Helicase subdomains and DUF283 are colored for visibility. (**D**) Details of interactions at the ATP binding pocket of dmDcr2, comparing the distance between Motif II and Motif VI for the apo enzyme and the enzyme in the presence of ATP and dsRNA. Green sphere is magnesium ion, a cofactor in Sf2 helicase ATP hydrolysis. (**E**) Structural alignment of predicted structures for AncD1D2 and AncD1$_{VERT}$ HEL-DUFs showing conformational differences in position of Hel1 and pincer subdomains and DUF283. Pincer and DUF293, left panel; Hel2 and Hel2i, middle panel; Hel1, right panel. Green and teal coloring represent AncD1$_{VERT}$ subdomains, and red and violet coloring represent AncD1D2 subdomains. Deuterostome-specific insert refers to a Hel2 insertion present in AncD1$_{DEUT}$ and AncD1$_{VERT}$. Structural predictions were performed with RosettaFold and AlphaFold2. pLDDT score: 81.76 for AncD1D2, 74.60 for AncD1$_{VERT}$. (**F**) Details of the interactions of the ATP binding pocket for AncD1D2 and AncD1$_{VERT}$, showing a wider cleft between Motif II and Motif VI for AncD1$_{VERT}$ (violet) compared to AncD1D2 (green). (**G**) RosettaFold predicted structures for AncD1$_{VERT}$ (transparent) showing sites of amino acid substitutions for both AncD1$_{VERT.1}$ and AncD1$_{VERT.7}$ marked in red, and amino acid substitutions unique to AncD1$_{VERT.7}$ marked in blue. ATP hydrolysis pocket is depicted.

properties. In AncD1$_{VERT}$, DUF283 (teal) is behind the helicase domain while AncD1D2's DUF283 (violet) caps the helicase cleft as in the closed conformation of dmDcr2 (*Figure 5E*, left panel). Hel2 (green) and Hel2i (teal) subdomains of AncD1$_{VERT}$ align closely with the corresponding subdomains in AncD1D2 (red and violet, respectively; *Figure 5E*, middle panel). However, AncD1$_{VERT}$'s Hel1 (green) leans away from the Hel2-Hel2i rigid body by 31.8° compared to AncD1D2's Hel1 subdomain (red; *Figure 5E*, right panel). This difference in conformation affects the ATP binding pocket which exists at the interface between Hel1 and Hel2, showing a wider distance between helicase motif II and motif VI in AncD1$_{VERT}$ compared to AncD1D2 (*Figure 5F*). This may explain why AncD1$_{VERT}$ is incapable of basal hydrolysis even when catalytic motifs are repaired in AncD1$_{VERT.1}$. Our studies indicate that non-catalytic motifs that affect conformation and helicase subdomain movement mediate the loss of ATPase function in vertebrate Dicer's helicase domain.

## Discussion

Phylogenetic tools have been used to analyze the evolution of the platform/PAZ domain in plant and animal Dicers (*Figure 1A*), shedding light on one source of functional diversity in eukaryotic Dicer (*Jia et al., 2017*; *Kidwell et al., 2014*; *Deddouche et al., 2008*). Here, we focused on evolution of Dicer's helicase domain in animals. An ATP-dependent helicase domain is important for Dicer's antiviral role in invertebrates such as dmDcr2 and ceDCR-1, while mammalian Dicer has not been observed to require ATP (*Welker et al., 2011*; *Donelick et al., 2020*; *Ashe et al., 2013*; *Li et al., 2016*). One explanation is that mammalian Dicer's helicase scaffold exists to stabilize the interaction of pre-miRNAs with the platform/PAZ domain during processing to mature miRNAs (*Liu et al., 2018*; *Gu et al., 2012*; *Zapletal et al., 2022*). Arthropods and nematodes are invertebrate ecdysozoan protostomes, and so far, these are the only two phyla where Dicer's helicase domain is known to be essential for antiviral defense. One wonders if this property is specific to Ecdysozoa or if it is more widespread among other bilaterian invertebrates like mollusks. The catalytic motif in this family of helicases is the DECH box, also known as motif II (*Luo et al., 2013*; *Fairman-Williams et al., 2010*). The DECH box is conserved between arthropods, nematodes, and mammals, but significant divergence in amino acid sequence of hsDcr, dmDcr2, and ceDCR-1 makes it challenging to answer these questions simply by analyzing amino acid variation. By performing APR, we generated evolutionary intermediates that revealed more subtle changes in amino acid variation and biochemical function, allowing insight into the biochemical properties of the ancient helicase domain and how these evolved to give rise to extant Dicer's roles in gene regulation and antiviral defense. Uncertainty in the single gene or protein phylogeny used for generating ancestral nodes and limitations in our ability to accurately predict the ancient amino sequence at deeper ancestral nodes are caveats of this approach. However, this is outweighed by the robust sequence-function analysis of Dicer's helicase domain enabled by APR that allowed interrogation of large sequence space that may not exist in any ancient or extant Dicers, as is the case for ML tree-species tree incongruent nodes like AncD1$_{ARTH/LOPH/DEUT}$ and AncD1$_{LOPH/DEUT}$.

### Ancestral animal Dicer possessed an active helicase domain

Our analysis revealed that AncD1D2, the common ancestor of dmDcr2, hsDcr, and ceDCR-1, retained the capability to hydrolyze ATP (*Figure 2A*). In addition, our phylogeny construction, performed with the helicase domain and DUF283, recapitulates the early gene duplication event reported previously in phylogenetic studies of full-length Dicers (*Figure 1D*; *Jia et al., 2017*; *Gao et al., 2014*). Plants and fungi have been reported to have Dicer or Dicer-like proteins with active helicase domains (*Kidwell et al., 2014*; *Wei et al., 2021*), so it stands to reason that early animal Dicer descended from an ancestral eukaryotic Dicer with an active helicase (*Shabalina and Koonin, 2008*). The ATP hydrolysis observed in our Dicer ancestors is predicted to be coupled to some motor function as observed in extant arthropod Dicer-2 (*Singh et al., 2021*). Future studies will determine if these constructs couple ATP hydrolysis to translocation and/or unwinding like dmDcr2 or to some other function like terminus discrimination.

## Ancient antiviral Dicer helicase function is retained in some invertebrates and lost in others

While the monophyletic arthropod Dicer-2 clade descends from $AncD2_{ARTH}$, in our ML tree (*Figure 1D*), other invertebrate Dicers cluster in unexpected positions in the Dicer-1 clade. Nematode HEL-DUFs diverge before the non-bilaterian cnidarians while mollusks cluster with deuterostomes and vertebrates. This phenomenon is likely due to the artifacts of incomplete lineage sorting and long branch attraction, but correlating the unusual phylogeny with existing genetic and biochemical data reveals some important trends in the evolution of innate immune defense mechanisms. The reduced ATPase observed in the ancestor of deuterostomes, $AncD1_{DEUT}$, and in the incongruent $AncD1_{LOPH/DEUT}$ (*Table 1*), suggests that some decline of helicase function occurred prior to deuterostome divergence from protostomes. This is further supported by the existence of RLR helicases and an interferon-like defense mechanism in mollusks (*Qiao et al., 2021*; *Lu et al., 2018*; *Huang et al., 2017*), the expansion of the miRNA repertoire in cephalopods (*Zolotarov et al., 2022*), and the observation that some extant mollusk Dicers have degenerate ATPase motifs (*Simakov et al., 2013*), implying that the antagonism between Dicer helicase function and the RLR-interferon axis observed in vertebrates predates deuterostome evolution. On the other hand, ceDCR-1 functions in synergy with a RLR homolog, DRH-1, in antiviral defense (*Ashe et al., 2013*; *Coffman et al., 2017*), indicating that this relationship is not universally antagonistic. Extant Dicers from ecdysozoan invertebrates like nematodes and arthropods have helicase-dependent antiviral function, but other invertebrates like the lophotrochozoan mollusk and deuterostome echinoderms may preferentially utilize a protein-based innate immune defense mechanism instead of Dicer's helicase domain for antiviral defense. Studying the modern Dicers from these organisms will shed light on how Dicer ATPase function has evolved in different invertebrates.

## Dicer ATPase function is lost entirely at the onset of vertebrate evolution

As animals evolved from deuterostomes ($AncD1_{DEUT}$) to vertebrates ($AncD1_{VERT}$), Dicer lost the ability to hydrolyze ATP entirely (*Figure 2C and D*). The loss of both intrinsic and dsRNA-stimulated ATPase activity between $AncD1_{DEUT}$ and $AncD1_{VERT}$ may be attributed to any number of evolutionary events. Whole genome duplication events that occurred at the onset of vertebrate evolution may have led to subfunctionalization of Dicer's helicase domain, as other antiviral sensors like RLRs and Toll-like receptors co-opted the role of sensing pathogen-associated molecular patterns (PAMPs; *Dehal and Boore, 2005*; *Liu et al., 2020*). Upon binding dsRNAs, these receptors trigger an enzyme cascade that ultimately produces interferon, a molecule that came into being at the onset of vertebrate evolution (*Secombes and Zou, 2017*). In support of this model, there are multiple examples of antagonism between the RNAi pathway and the RLR signaling pathway in mammals (*van der Veen et al., 2018*; *Seo et al., 2013*; *Gurung et al., 2021*; *Witteveldt et al., 2019*). It is also likely that Dicer helicase function started to decline before the advent of deuterostomes as different components of the RLR-interferon axis appeared earlier in animal evolution (*Qiao et al., 2021*; *Huang et al., 2017*). Separately, the expansion and specialization of pre-miRNA substrates requiring Dicer's platform/PAZ domain may have contributed to, or benefited from, a loss of helicase function (*Berezikov, 2011*). This model is further supported by our dsRNA binding data where weaker dsRNA binding by the deuterostome and vertebrate helicase domains suggests that RLRs assumed cytosolic viral dsRNA recognition and the platform/PAZ domain became the dominant endogenous dsRNA recognition motif (*Figure 3*, *Table 2*). The order in which these events would have occurred is unclear, but if this model is true, it raises new questions about why Dicer's helicase function could not co-exist with a protein-based innate immune system and miRNA expansion. Could vertebrates maintain both modes of antiviral defense if loss of function in Dicer's helicase domain was reversed? Further studies on the selection pressures exerted on Dicer in different species by different RNA substrates are required to answer these questions.

## ATP and dsRNA binding are limiting factors in vertebrate Dicer helicase function

The role of hsDcr in antiviral defense is controversial (*Li et al., 2016*; *Qiu et al., 2017*; *tenOever, 2017*; *Maillard et al., 2013*; *Maillard et al., 2016*; *Parameswaran et al., 2010*). The current

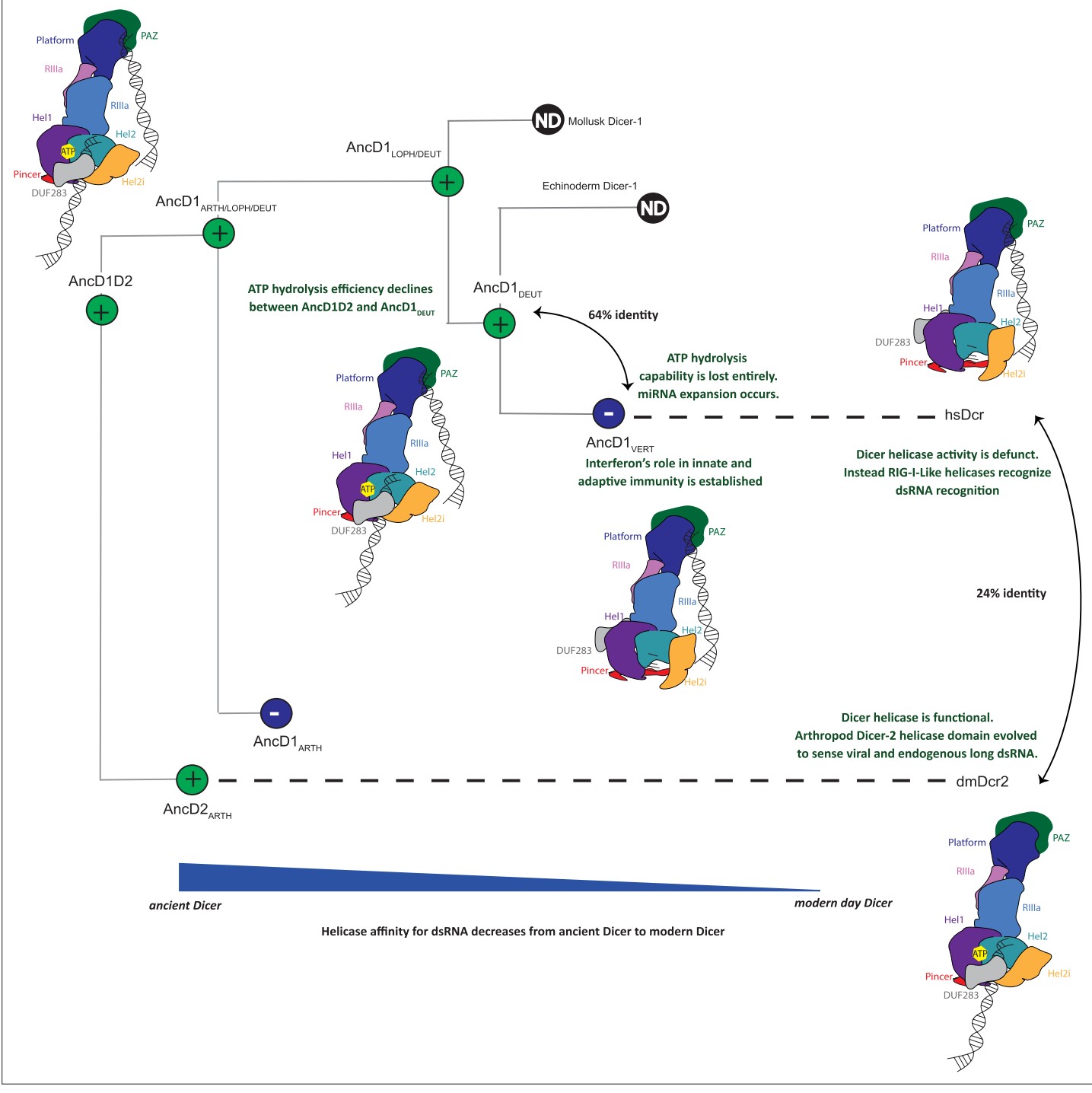

**Figure 6.** Model of metazoan Dicer evolution showing transition from a two-site dsRNA binding in ancestral Dicer to a 1-site dsRNA binding state in extant vertebrate and arthropod Dicers. Early animals possessed one promiscuous Dicer enzyme capable of using both platform/PAZ and helicase domains for dsRNA recognition. After gene duplication, arthropod Dicer-2's helicase domain becomes specialized for viral and endogenous long dsRNA processing and becomes the primary site of dsRNA binding. Deuterostome Dicer-1 may have retained two-site dsRNA recognition as helicase function declined, but at the onset of vertebrate evolution, Dicer-1 loses helicase function entirely and exclusively uses the platform/PAZ domain for dsRNA recognition.

consensus is that hsDcr is more relevant for antiviral defense in stem cells, while RLRs and interferon signaling predominate in somatic cells (*van der Veen et al., 2018*; *Witteveldt et al., 2019*). HsDcr's helicase domain is, however, not involved in stem-cell specific antiviral function, and in fact, cleavage of viral or endogenous long dsRNA is improved when the helicase domain is truncated or removed (*Poirier et al., 2021*; *Flemr et al., 2013*). This suggests that hsDcr's helicase domain is incapable of coupling dsRNA translocation to ATP hydrolysis. Our dsRNA binding data reinforce this observation: dsRNA binding is significantly worse in AncD1$_{VERT}$ than it is for AncD1D2, suggesting that vertebrates in general do not use Dicer's helicase domain for antiviral defense (*Figure 3*). Reinforcing this model, cryo-EM structures of hsDcr report an open conformation for hsDcr's helicase domain even in the presence of dsRNA (*Figure 1C*), and cleavage of long dsRNAs by hsDcr is mediated by direct binding to the platform/PAZ domain with no requirement for ATP (*Liu et al., 2018*; *Park et al., 2011*).

Canonical SF2 helicase ATP binding/hydrolysis motifs, like the eponymous 'DECH' box, are conserved between hsDcr and dmDcr2, but outside these motifs, the primary sequence varies significantly (*Sinha et al., 2018*). Using sequence- and structure-based alignments of our ancestral HEL-DUF constructs, we identified candidate historical substitutions outside the catalytic motifs (*Figure 4—figure supplement 2*) that caused the loss of intrinsic and BLT dsRNA-stimulated ATP hydrolysis in AncD1$_{VERT}$ (*Figure 5G*). We created AncD1$_{VERT.7}$, a construct with partial rescue of basal and BLT dsRNA-stimulated ATP hydrolysis activity, with efficiency an order of magnitude lower than AncD1$_{DEUT}$ and AncD1D2. Michaelis-Menten analysis indicated that the limiting factor in our rescue construct was low affinity for ATP, indicated by the $K_M$ value. This indicates that AncD1$_{VERT}$ does not hydrolyze ATP, despite the conservation of the catalytic 'DECH' box because several motifs distant from the ATPase catalytic site are responsible for loss of ATP hydrolysis capability. In AncD1$_{VERT}$ and hsDcr, these residues likely stabilize the open conformation of the helicase domain, making the formation of the ATP hydrolysis pocket at the interface between Hel1 and Hel2 very unlikely (*Figure 4—figure supplement 2*, *Figure 5*). In summary, our analyses indicate that loss of ATPase function in vertebrate Dicer, and consequently hsDcr, is caused by a set of mutations that limit formation of the ATPase pocket, as well as dsRNA binding and the conformational changes it would ordinarily trigger. Further engineering of AncD1$_{VERT}$ is required to create a version of vertebrate Dicer that hydrolyses ATP more efficiently and couples this hydrolysis to improved viral siRNA production in the context of the full-length enzyme.

In our favorite model for Dicer evolution (*Figure 6*), the full-length ancestral animal Dicer was capable of binding dsRNAs at two sites: the platform/PAZ domain for pre-miRNA and HEL-DUF for long endogenous or viral dsRNA (*Jia et al., 2017*). This Dicer was probably more promiscuous and possessed the ability to translocate dsRNA from helicase to platform/PAZ. After duplication, arthropod Dicer-2 evolved to a one-site mechanism where the HEL-DUF became the primary site of dsRNA recognition. This idea is supported by previous work showing that compared to the Dicer-1 clade, the platform/PAZ domain in arthropod Dicer-2 lost affinity for dsRNA (*Jia et al., 2017*) and analyses of *Drosophila* species that revealed evidence of positive selection in Dicer-2 HEL-DUF domains (*Kolaczkowski et al., 2011*). Thus, dsRNA and ATP binding to dmDcr2's helicase domain drive conformational changes in the entire enzyme and leads to translocation of dsRNA to the RNaseIII domain for cleavage (*Su et al., 2022*; *Singh et al., 2021*). Drosha appears early in animal evolution and may have contributed to Dicer-1 specialization by performing processing of pri-miRNAs, a step performed by Dicer's helicase domain in plants (*Moran et al., 2017*). Dicer-1 underwent a series of evolutionary changes between AncD1D2 and AncD1$_{DEUT}$, culminating in vertebrate Dicer's helicase domain losing affinity for both dsRNA and ATP. This decline in helicase function coincides with the appearance of the antiviral RLRs and interferon-like cytokines and may have facilitated subsequent miRNA expansion in vertebrates and mollusks. Thus, vertebrate Dicer works with a one-site mechanism where the platform/PAZ domain is the predominant binding site for all its dsRNAs substrates. Further studies exploring how Dicer-1 enzymes from other invertebrates, like mollusks and echinoderms, process dsRNAs will provide a clearer picture of how different Dicer domains evolved to contribute to different RNAi pathways.

## Materials and methods

**Key resources table**

| Reagent type (species) or resource | Designation | Source or reference | Identifiers | Additional information |
|---|---|---|---|---|
| Recombinant DNA reagent | pFastBac-OSF | Thermo Fisher Scientific | Cat# 10360014 | Modified in-house to add OSF tag |
| Cell line (*Spodoptera frugiperda*) | Sf9 | Expression Systems | Cat# 94–001 S | Suspension insect cells |
| Strain and strain background (*Escherichia coli*) | DH10Bac | Thermo Fisher Scientific | Cat #10361012 | Max Efficiency Competent Cells |
| Antibody | Anti-gp64-PE (mouse, monoclonal) | Expression Systems | Cat# 97–101 | Baculovirus Titering Kit |
| Chemical compound and drug | Cellfectin II | Thermo Fisher Scientific | Cat# 10362100 | Transfection reagent |
| Software and algorithm | RAXML-NG | RAXML-NG | RRID:SCR_022066 | |
| Sequence-based reagent | 42-nucleotide sense RNA | Integrated DNA Technologies (IDT) | Single-stranded RNA | GGGAAGCUCAGAAUA UUGCACAAGUAGAGC UUCUCGAUCCCC |
| Sequence-based reagent | 42-nucleotide BLUNT antisense RNA | IDT | Single-stranded RNA | GGGGAUCGAGAAGCU CUACUUGUGCAAUAU UCUGAGCUUCCC |
| Sequence-based reagent | 42-nucleotide 3'overhang antisense RNA | IDT | Single-stranded RNA | GGAUCGAGAAGCUCUA CUUGUGCAAUAUUCUG AGCUUCCCGG |

## Phylogenetics and APR

Annotated Dicer protein sequences were retrieved from the NCBI database using taxa from each of the main animal phyla as queries for the BLAST algorithm (*McGinnis and Madden, 2004*). Representative protein sequences from each metazoan phylum were used as search templates to retrieve a wide range of Dicer orthologs and paralogs with fungus Dicers used as the outgroup to root the animal clade. Protein sequences were clustered using CD-HIT with an identity threshold of 95% and representative sequences aligned initially with MAFFT (*Li and Godzik, 2006*; *Katoh and Standley, 2013*). Initial MSA was used to assess and visualize helicase domain and DUF283 boundaries as defined by the Conserved Domain Database (*Marchler-Bauer et al., 2007*). All downstream analysis was performed on the helicase and DUF283 referred to as HEL-DUF. Large gaps were manually deleted from the initial HEL-DUF alignment, and PRANK was used to generate a new alignment (*Löytynoja and Goldman, 2008*). Manual curation was carried out to remove species-specific indels and exclude sequences missing conserved parts of the helicase domain or DUF283. Model selection was performed on the resulting MSA using ProtTest 3.4.2, producing JTT +G + F as the best fit evolutionary model using the Akaike Information Criteria (*Darriba et al., 2011*).

RAXML-NG v 1.0.1 was used to infer the ML phylogeny using the best fit evolutionary model, with eight rate categories in a gamma distribution to model among-site rate variation (*Kozlov et al., 2019*). Transfer Bootstrap was used to calculate statistical support for the ancestral nodes with fungal Dicers used as the outgroup for rooting the tree. Ancestral state reconstruction in RAXML-NG using the ML tree and the JTT model (*Kozlov et al., 2019*; *Lemoine et al., 2018*). Because an especially gappy alignment was produced as a result of using PRANK which models every unique insertion as a separate evolutionary event (*Löytynoja and Goldman, 2008*; *Vialle et al., 2018*; *Löytynoja, 2013*), the input protein MSA was converted to a presence-absence alignment to model the indels in the alignment, and this matrix was used to perform APR with the ML phylogeny and the BINCAT model in RAXML-NG (*Jia et al., 2017*; *Aadland et al., 2018*). Overlapping the protein and binary sequence ancestral reconstructions allowed the identification of spurious indels in ancestral sequences by eliminating low-frequency insertions that were missed during manual curation.

## Cloning and protein expression

DNA sequences coding for select ancestral protein sequences were codon-optimized for expression in *Spodoptera frugiperda* (Sf9) insect cells using Integrated DNA Technologies' (IDT) codon optimization tool. cDNA sequences were synthesized by IDT and subcloned into a modified pFastBac plasmid containing 2×-Strep Flag tag. Plasmids were transformed into Dh10Bac *Escherichia coli* cells

to generate bacmids, which were transfected into Sf9 cells to produce baculovirus vectors for protein expression (*Sinha and Bass, 2017*). Baculovirus titer was quantified using flow cytometry (*Mulvania et al., 2004*). Ancestral HEL-DUF constructs were purified using Strep-Tactin affinity chromatography, heparin chromatography or ion exchange chromatography, and size exclusion chromatography. All ancestral constructs eluted as monomers except ANCD1$_{VERT}$, which eluted as a mixture of monomers and dimers. Purified constructs were identified using mass spectrometry at UC Davis Proteomics Core Facility.

## dsRNA preparation

42 nt ssRNAs were chemically synthesized by University of Utah DNA/RNA Synthesis Core or IDT. Equimolar amounts of ssRNAs were annealed in annealing buffer (50 mM TRIS pH 8.0, 20 mM KCl) by placing the reaction on a heat block (95°C) and slow cooling ≥2 hr (*Sinha and Bass, 2017*). dsRNAs were gel purified after 8% polyacrylamide native PAGE and quantified using a Nanodrop.

## RNA sequences

42-nt sense RNA:
5'-GGGAAGCUCAGAAUAUUGCACAAGUAGAGCUUCUCGAUCCCC-3'
42-nt antisense BLUNT RNA:
5'-GGGGAUCGAGAAGCUCUACUUGUGCAAUAUUCUGAGCUUCCC-3'
42-nt antisense 3'OVR RNA:
5'-GGAUCGAGAAGCUCUACUUGUGCAAUAUUCUGAGCUUCCCGG-3'

## ATP hydrolysis assays

Reactions were performed at 37°C in 65 μL reaction mixtures containing cleavage assay buffer (25 mM TRIS pH 8.0, 100 mM KCl, 10 mM MgCl$_2$, and 1 mM TCEP) for the times indicated, with 200 nM ancestral protein and 400 nM 42 BLT or 3'ovr dsRNA, in the presence of 100 μM ATP-MgOAc with [α-$^{32}$P] ATP (3000 Ci/mmol, 100 nM) spiked in to monitor hydrolysis. Protein was preincubated at 37°C for 3 min prior to the addition to reaction mix. Reactions were started by the addition of protein to reaction mix containing ATP and dsRNA. 2 μL of reaction were removed at indicated times, quenched by addition of 2 μL of 500 mM EDTA, spotted (3 μL) onto 20×20 cm PEI-cellulose plates (Cel 300 PEI/ UV 254 TLC Plates 20x20, Machery-Nagel, Ref 801063), and chromatographed with 0.75 M KH$_2$PO$_4$ (adjusted to pH 3.3 with H$_3$PO$_4$) until solvent front reached the top of the plate. Plates were dried, visualized on a PhosphorImager screen (Molecular Dynamics), and quantified using ImageQuant software.

Quantification of ATP hydrolysis assays for *Table 1* was done by fitting the data into a two-phase exponent, with the first phase modeled as a linear reaction between time 0 and 2.5 min. The first phase is considered to be a transient zero order reaction where the rate constant k is equal to the velocity of the reaction, which is the slope of ADP produced/ATP consumed (y) per minute (t). In reality, this rapid first phase probably ended before 2.5 min, but we are limited by the nature of manually mixed assays, as opposed to stopped flow assays where mixing and signal collection can be done on the timescale of seconds.

$$y=kt + \text{intercept}$$

Data for the second phase were fit to the pseudo-first order equation $y=y_o + A \times (1-e^{-kt})$; where y=product formed (ADP in μM); A=amplitude of the rate curve, $y_o$ = baseline, k=pseudo-first-order rate constant = $k_{obs}$; t=time. Data points are mean ± SD (n≥3).

For the Michaelis-Menten ATP hydrolysis assays, varying amounts of ATP-MgOAc with (α-$^{32}$P) ATP (3000 Ci/mmol, 50 nM) were incubated with the indicated protein concentrations, and the velocity of the steady-state reaction was calculated using a linear regression:

ADP produced (μM)=velocity (μM/min) × time (min).

The velocity recorded for each starting ATP concentration was fit into the Michaelis-Menten equation:

Velocity = $V_{max} * X/(K_M +X)$, where $V_{max}$ is the maximum enzyme velocity (μM/min), X is the ATP concentration (mM), and $K_M$ is the Michaelis-Menten constant.

The turnover number, $k_{cat}$, was calculated by dividing $V_{max}$ by the total enzyme concentration. GraphPad Prism version 9 was used for curve-fitting analysis.

## Gel shift mobility assays

Gel mobility shift assays were performed with 20pM 42-basepair BLT or 3'ovr dsRNA, with sense strand labeled with $^{32}$P at the 5' terminus. Labeled dsRNA was incubated and allowed to reach equilibrium (30 min, 4°C) with HEL-DUF construct in the presence and absence of 5 mM ATP-MgOAc$_2$, in binding buffer (25 mM TRIS pH 8.0, 100 mM KCl, 10 mM MgCl$_2$, 10% [vol/vol] glycerol and 1 mM TCEP); final reaction volume, 20 μL. Ancestral HEL-DUF protein was serially diluted in binding buffer before addition to binding reaction. Reactions were stopped by loading directly onto a 5% polyacrylamide (19:1 acrylamide/bisacrylamide) native gel running at 200 V at 4°C, in 0.5× Tris/Borate/EDTA running buffer. The gel was pre-run (30 min) before loading samples. Gels were electrophoresed (2 hr) to resolve HEL-DUF-bound dsRNA from free dsRNA, dried (80°C, 1 hr), and exposed overnight to a Molecular Dynamics Storage Phosphor Screen. Radioactivity signal was visualized on a Typhoon PhosphorImager (GE Healthcare LifeSciences) in the linear dynamic range of the instrument and quantified using ImageQuant version 8 software.

Radioactivity in gels corresponding to dsRNA$_{total}$ and dsRNA$_{free}$ was quantified to determine the fraction bound. Fraction bound = $1 - (dsRNA_{free}/dsRNA_{total})$. All dsRNA that migrated through the gel more slowly than dsRNA$_{free}$ was considered as bound. To determine $K_d$ values, binding isotherms were fit using the Hill formalism, where fraction bound = $1/(1 + [K_d^n/[P]^n])$; $K_d$ = equilibrium dissociation constant, n=Hill coefficient, and [P]=protein concentration (*Jarmoskaite et al., 2020*). GraphPad Prism version 9 was used for curve-fitting analysis.

## Acknowledgements

We thank Alesia McKeown, Nels Elde, Paul Sigala, Peter Shen, Demian Cazalla, Tyler Starr and Thomas Koch, as well as members of the Bass Lab for helpful discussions and feedback. We thank Claudia Consalvo for help with assay design and assistance with flow cytometry. We thank James Marvin for supervision of flow cytometry procedure, Ryan Andrews for help with figure design, and Michelle Salemi (UC Davis Proteomics Core) for assistance with protein mass spectrometry. AA was supported by the 3i Initiative Graduate Fellowship at the University of Utah. DNA synthesis and flow cytometry was performed at the University of Utah Core Facilities. This work was supported by funding to BLB from the National Institute of General Medical Sciences (R35GM141262) and the National Cancer Institute of the National Institutes of Health (R01CA260414).

## Additional information

### Funding

| Funder | Grant reference number | Author |
| --- | --- | --- |
| School of Medicine | 3I Graduate Student Fellowship | Adedeji M Aderounmu |
| National Institute of General Medical Sciences | R35GM141262 | Brenda L Bass |
| National Cancer Institute | R01CA260414 | Brenda L Bass |

The funders had no role in study design, data collection and interpretation, or the decision to submit the work for publication.

### Author contributions

Adedeji M Aderounmu, Conceptualization, Data curation, Software, Formal analysis, Investigation, Visualization, Methodology, Writing - original draft, Writing - review and editing; P Joseph Aruscavage, Investigation, Methodology; Bryan Kolaczkowski, Data curation, Software, Formal analysis, Supervision, Validation, Methodology; Brenda L Bass, Conceptualization, Resources, Data curation,

Software, Formal analysis, Supervision, Funding acquisition, Validation, Investigation, Visualization, Methodology, Writing - original draft, Project administration, Writing - review and editing

### Author ORCIDs
Adedeji M Aderounmu http://orcid.org/0000-0002-9981-7533
Brenda L Bass http://orcid.org/0000-0003-1728-2254

### Decision letter and Author response
Decision letter https://doi.org/10.7554/eLife.85120.sa1
Author response https://doi.org/10.7554/eLife.85120.sa2

## Additional files

### Supplementary files
• Supplementary file 1. Fasta file containing amino acid sequences of ancestrally reconstructed proteins and engineered protein constructs.

• Supplementary file 2. Fasta file containing multiple sequence alignment file used as input for phylogeny construction and ancestral protein reconstruction. Protein accession numbers from NCBI.

• Supplementary file 3. Text file containing the reconstructed ancestral states for all ancestral nodes in the maximum likelihood phylogenetic tree. Reconstructions represent primary amino acid sequences prior to removal of low-probability insertions using binary states generated from the BIN model in RAXML-NG.

• Supplementary file 4. Text file containing the reconstructed ancestral states for all ancestral nodes in the maximum likelihood phylogenetic tree, using the multiple sequence alignment in the form of an absence-presence matrix and the BIN model in RAXML-NG for ancestral protein reconstruction. One (1) represent the presence of amino acid residues, and zero (0) represents the absence. Protein accession numbers from NCBI.

• Supplementary file 5. Newick file of maximum likelihood metazoan Dicer HEL-DUF phylogeny used for ancestral reconstruction (shown in *Figure 1* and *Figure 1—figure supplement 1*).

• Supplementary file 6. Newick file of maximum likelihood phylogeny with bilaterian species constrained (shown in *Figure 1—figure supplement 2A*). Protein accession numbers from NCBI.

• Supplementary file 7. Text files containing python scripts used to convert the amino acid multiple sequence alignment to binary alignment, overlay both alignments, and remove amino acids that correspond with absence (0) in the binary alignment.

• MDAR checklist

### Data availability
All data generated or analyzed during this study are included in the manuscript and supporting files. Source data files have been provided for all gels displayed in Figures 2, 3 and 4. Phylogenetic and ancestral protein reconstruction data are provided as supplementary files.

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
