## [Editor Report]

This is a valuable paper describing an attempt to reconstruct the evolution of Dicer. Using ancestral reconstruction approaches, the authors carefully examine the biochem ical characteristics of reconstructed proteins at various junction points in the animal lineage. The authors provide solid evidence that the deepest ancestrally reconstructed protein has double-stranded RNA stimulated ATPase activity and that this characteristic was lost along the vertebrate lineage. This paper will be of interest to scientists in the RNA-protein interaction and protein evolution fields.

---

## [Decision Letter]

**Decision letter after peer review:**

Thank you for submitting your article "Ancestral protein reconstruction reveals evolutionary events governing variation in Dicer helicase function" for consideration by *eLife*. Your article has been reviewed by 2 peer reviewers, and the evaluation has been overseen by a Reviewing Editor and Volker Dötsch as the Senior Editor. The reviewers have opted to remain anonymous.

The reviewers have discussed their reviews with one another, and the Reviewing Editor has drafted this to help you prepare a revised submission. As you see from the comments some clarifications would improve the manuscript.

Essential revisions:

1) Line 111: Please explain well early in the article what AncD1D2 represents, I think this is a very important point. From some statements it seems it could be the last common Dicer before the split of Dicer-1 and Dicer-2 lineages, which would indicate it's the ancestral bilaterian Dicer, but from the phylogenetic tree in Figure 1C it seems it is some Eumetazoan, perhaps Bilaterian, Dicer after of Placozoan ancestor separation. Figure 1 supplement 1 suggests it is ancestral Dicer common for Cnidarians and Bilaterians. And the alignment of species in the supplement contains also Dicers from Poriphera. The information where this Dicer fits into the evolution is important and should be explained well.

2) line 332 – "Changes in Dicer helicase domain's conformation …" structural data presented in this part of the discussion should be included in the results instead of appearing in the middle of the discussion.

3) The model (Figure 6): Available evidence (discussed also in the public review). There is evidence that the model does not correspond well to available data.

4) The paper would be strengthened if the authors would consider alternative explanations to the questions that are currently not answerable. For example, the authors speculate on the order of several evolutionary events, but it is worth pointing out that the order has not been resolved and offer alternative scenarios. They are certainly free to express their preference, but it would be wise to explicitly state the alternatives,

*Reviewer #1 (Recommendations for the authors):*

The paper would be strengthened if the authors would consider alternative explanations to the questions that are currently not answerable. For example, the authors speculate on the order of several evolutionary events, but it is worth pointing out that the order has not been resolved and offer alternative scenarios. They are certainly free to express their preference, but it would be wise to explicitly state the alternatives,

*Reviewer #2 (Recommendations for the authors):*

The authors took an original approach and produced interesting data. Biochemical analyses are of high quality and regarding the reconstructed hypothetical ancestral sequences, they provide remarkable information and can be accepted as reasonable approximations to the ancestral states. While I am curious how would the helicases perform in the context of full-length proteins, I think the work covers the topic well and does not need to be further expanded. My major reservations concern text clarity and author's interpretations, some of which I mentioned in the public review, so pardon some redundancy, please. Below I will make additional comments:

1) Line 111: Please explain well early in the article what AncD1D2 represents, I think this is a very important point. From some statements it seems it could be the last common Dicer before the split of Dicer-1 and Dicer-2 lineages, which would indicate it's the ancestral bilaterian Dicer, but from the phylogenetic tree in Figure 1C it seems it is some Eumetazoan, perhaps Bilaterian, Dicer after of Placozoan ancestor separation. Figure 1 supplement 1 suggests it is ancestral Dicer common for Cnidarians and Bilaterians. And the alignment of species in the supplement contains also Dicers from Poriphera. The information where this Dicer fits into the evolution is important and should be explained well.

2) line 332 – "Changes in Dicer helicase domain's conformation …" structural data presented in this part of the discussion should be included in the results instead of appearing in the middle of the discussion.

3) The model (Figure 6): Available evidence. There is evidence that the model does not correspond well to available data.

---

## [Author Response]

Essential revisions:1) Line 111: Please explain well early in the article what AncD1D2 represents, I think this is a very important point. From some statements it seems it could be the last common Dicer before the split of Dicer-1 and Dicer-2 lineages, which would indicate it's the ancestral bilaterian Dicer, but from the phylogenetic tree in Figure 1C it seems it is some Eumetazoan, perhaps Bilaterian, Dicer after of Placozoan ancestor separation. Figure 1 supplement 1 suggests it is ancestral Dicer common for Cnidarians and Bilaterians. And the alignment of species in the supplement contains also Dicers from Poriphera. The information where this Dicer fits into the evolution is important and should be explained well.

We thank the reviewer for bringing this issue to our attention. Early in the Results section, as soon as we present our maximum-likelihood tree, we now include a discussion of the complexities in identifying AncD1D2’s exact position in metazoan evolution (page 6, line 124, beginning with “Importantly, we observed multiple instances where our ML tree was not congruent with the species tree…”). Our hypothetical maximum likelihood tree (Figure 1D) is constructed from sequences corresponding to Dicer’s helicase and DUF283 domains and does not contain enough phylogenetic signal to resolve finer details of early metazoan evolutionary events surrounding the divergence of non-bilaterians: Porifera, Ctenophora, Cnidaria and Placozoa. In our tree, Cnidaria even diverges later than the Nematode bilaterian branch, reflecting that our reported phylogeny does not always match consensus species relationships. This means we cannot pinpoint AncD1D2’s exact position with certainty. While we do not intend to overinterpret the evolutionary trends from these hypothetical ancestral constructs, we believe the functional differences in biochemical activity are meaningful and correspond to big-picture changes over evolutionary time. In our analyses, AncD1D2 corresponds to some early metazoan ancestor that existed before the divergence of bilaterians. In support of this interpretation, when the phylogeny is constrained such that the bilaterian branches match the consensus species tree (Figure 1—figure supplement 2A) the version of AncD1D2 from this tree is ancestral to the bilaterian ancestor AncD1_BILAT_, which we have now labeled, and retains 95% identity to the version of AncD1D2 constructed from the maximum likelihood phylogeny but only ~68% identity to AncD1_BILAT_ amino acid sequence.

2) line 332 – "Changes in Dicer helicase domain's conformation …" structural data presented in this part of the discussion should be included in the results instead of appearing in the middle of the discussion.

We have moved this section from the Discussion section to the Results section in the revised manuscript. (Page 13, line 313)

3) The model (Figure 6): Available evidence (discussed also in the public review). There is evidence that the model does not correspond well to available data.

We agree with the reviewer that the existing data from mollusks and other invertebrates suggests that we have focused too heavily on the loss of helicase function in the deuterostome-to-vertebrate transition, as opposed to the more subtle losses of helicase function observed prior to AncD1_DEUT_. We have now added a new section to the Discussion focused on the possible competition between a protein-based antiviral mechanism and the antiviral Dicer helicase-dependent RNAi mechanism in invertebrates, and on the changes between AncD1D2 and AncD1_DEUT_ (page 16, line 384, "Ancient antiviral Dicer helicase function is retained in some invertebrates and lost in others"). We have also modified the text included on the model (Figure 6) and in the Discussion (page 21, lines 511-518, starting with “Drosha appears early in animal evolution and…”) to reflect the possibility that an interferon-like antiviral defense mechanism existed prior to vertebrate evolution and may have contributed to the decline of Dicer’s helicase domain.

4) The paper would be strengthened if the authors would consider alternative explanations to the questions that are currently not answerable. For example, the authors speculate on the order of several evolutionary events, but it is worth pointing out that the order has not been resolved and offer alternative scenarios. They are certainly free to express their preference, but it would be wise to explicitly state the alternatives,

We agree with the reviewer that our manuscript would be improved by including discussion of alternative hypotheses. We have added alternative explanations and models for Dicer duplication (page 6, lines 128-134, beginning with “We observed several incongruences…”), and the possibility of competition between different antiviral pathways as well as the potential involvement of miRNA expansion (page 16, section beginning on line 384; page 18, lines 418-422, line beginning with “It is also likely that Dicer helicase function…”).

Reviewer #1 (Recommendations for the authors):The paper would be strengthened if the authors would consider alternative explanations to the questions that are currently not answerable. For example, the authors speculate on the order of several evolutionary events, but it is worth pointing out that the order has not been resolved and offer alternative scenarios. They are certainly free to express their preference, but it would be wise to explicitly state the alternatives,

Addressed under public review and Essential revision 4.

Reviewer #2 (Recommendations for the authors):The authors took an original approach and produced interesting data. Biochemical analyses are of high quality and regarding the reconstructed hypothetical ancestral sequences, they provide remarkable information and can be accepted as reasonable approximations to the ancestral states. While I am curious how would the helicases perform in the context of full-length proteins, I think the work covers the topic well and does not need to be further expanded. My major reservations concern text clarity and author's interpretations, some of which I mentioned in the public review, so pardon some redundancy, please. Below I will make additional comments:1) Line 111: Please explain well early in the article what AncD1D2 represents, I think this is a very important point. From some statements it seems it could be the last common Dicer before the split of Dicer-1 and Dicer-2 lineages, which would indicate it's the ancestral bilaterian Dicer, but from the phylogenetic tree in Figure 1C it seems it is some Eumetazoan, perhaps Bilaterian, Dicer after of Placozoan ancestor separation. Figure 1 supplement 1 suggests it is ancestral Dicer common for Cnidarians and Bilaterians. And the alignment of species in the supplement contains also Dicers from Poriphera. The information where this Dicer fits into the evolution is important and should be explained well.

Addressed in Essential revisions point 1.

2) line 332 – "Changes in Dicer helicase domain's conformation …" structural data presented in this part of the discussion should be included in the results instead of appearing in the middle of the discussion.

Addressed in Essential revisions point 2.

3) The model (Figure 6): Available evidence. There is evidence that the model does not correspond well to available data.

Addressed in Essential revisions point 3.